# Expanded vacuum-stable gels for multiplexed high-resolution spatial histopathology

Yunhao Bai[1,2], Bokai Zhu[1,3], John-Paul Oliveria[4,5], Bryan J. Cannon ● [1], Dorien Feyaerts ● [6], Marc Bosse ● [1], Kausalia Vijayaragavan[1], Noah F. Greenwald ● [1], Darci Phillips ● [1], Christian M. Schürch ● [1,7], Samuel M. Naik ● [8], Edward A. Ganio[6], Brice Gaudilliere ● [6], Scott J. Rodig[8], Michael B. Miller ● [8,9,10,11], Michael Angelo ● [1], Sean C. Bendall ● [1], Xavier Rovira-Clavé ● [1,3,14] ✉, Garry P. Nolan ● [1,14] ✉ & Sizun Jiang ● [1,11,12,13,14] ✉

Cellular organization and functions encompass multiple scales in vivo. Emerging high-plex imaging technologies are limited in resolving subcellular biomolecular features. Expansion Microscopy (ExM) and related techniques physically expand samples for enhanced spatial resolution, but are challenging to be combined with high-plex imaging technologies to enable integrative multiscaled tissue biology insights. Here, we introduce Expand and comPRESS hydrOgels (ExPRESSO), an ExM framework that allows high-plex protein staining, physical expansion, and removal of water, while retaining the lateral tissue expansion. We demonstrate ExPRESSO imaging of archival clinical tissue samples on Multiplexed Ion Beam Imaging and Imaging Mass Cytometry platforms, with detection capabilities of > 40 markers. Application of ExPRESSO on archival human lymphoid and brain tissues resolved tissue architecture at the subcellular level, particularly that of the blood-brain barrier. ExPRESSO hence provides a platform for extending the analysis compatibility of hydrogel-expanded biospecimens to mass spectrometry, with minimal modifications to protocols and instrumentation.

Microenvironments both within and around cells can determine cellular identity, function, and fate. Fluorescence microscopy is the de facto choice for biomolecular imaging at the mesoscale, but despite remarkable utilities, it continues to be hindered by physical constraints including the diffraction limit of light, fluorophore spectral overlap, and tissue autofluorescence. Overcoming these limitations will enable new frameworks to contextualize distinct biomolecules and their associated functions at the mesoscale, within their natural environments.

[1]Department of Pathology, Stanford University, Stanford, CA, USA. [2]Department of Chemistry, Stanford University, Stanford, CA, USA. [3]Department of Microbiology and Immunology, Stanford University, Stanford, CA, USA. [4]Department of Translational Medicine, Genentech, Inc., South San Francisco, CA, USA. [5]Department of Medicine, McMaster University, Hamilton, ON, Canada. [6]Department of Anesthesiology, Perioperative and Pain Medicine, Stanford University, Stanford, CA, USA. [7]Department of Pathology and Neuropathology, University Hospital and Comprehensive Cancer Center Tübingen, Tübingen, Germany. [8]Department of Pathology, Brigham and Women's Hospital, Harvard Medical School, Boston, MA, USA. [9]Department of Neurology, Brigham and Women's Hospital, Harvard Medical School, Boston, MA, USA. [10]Division of Genetics and Genomics, Department of Pediatrics, Boston Children's Hospital, Boston, MA, USA. [11]Broad Institute of MIT and Harvard, Cambridge, MA, USA. [12]Center for Virology and Vaccine Research, Beth Israel Deaconess Medical Center, Boston, MA, USA. [13]Department of Pathology, Dana Farber Cancer Institute, Boston, MA, USA. [14]These authors jointly supervised this work: Xavier Rovira-Clavé, Garry P. Nolan, Sizun Jiang. ✉e-mail: x.rovira.c@gmail.com; gpnolan@stanford.edu; sjiang3@bidmc.harvard.edu

Super-resolution microscopy methods circumvent the diffraction limit of light[1–3], but, unlike conventional fluorescence microscopy, these methods require highly specialized chemical probes, microscopy equipment, and experimental conditions. An alternative approach aimed at democratizing super-resolution imaging is to physically enlarge the sample, rather than improving instrumentation capabilities. Boyden and colleagues pioneered this concept, termed Expansion Microscopy (ExM)[4,5]. In ExM, an acrylate-based polymer mesh is interlaced into biological specimens prior to physical expansion induced by the replacement of a high salt buffer with water. The initial approach enabled an approximate fourfold expansion in the lateral and axial directions while diminishing autofluorescence[4]. Modifications of the original polymer chemistry enabled even larger fold expansions[6–9], multiplexed whole organ imaging[10], analysis of formalin-fixed paraffin-embedded (FFPE) archival pathological specimens[11], and of subcellular components[12,13].

Many current ExM-derived methods are dependent on fluorescence-based microscopy and thus share many of the same limitations. The most restrictive is the spectral overlap between fluorophores, which significantly limits multiplexing capabilities. Iterative multiplexing strategies, including CODEX[14,15] and CyCIF[16], are not easily compatible with ExM due to the chemical treatments involved in probe or antibody stripping. Although initial efforts towards multiplexed imaging within expanded tissues have been encouraging[10,17,18], scalability remains a challenge.

Mass spectrometry imaging (MSI), an alternative to fluorescence-based multiplexed imaging that images mass-tagged or intrinsic biomolecules, is capable of simultaneous acquisition of tens to hundreds of parameters[19–26]. Recent technical breakthroughs have allowed for antibody-based spatial proteomics analyses of tissues with MSI down to the single-cell level, including two alternative tagging and visualization methods termed Multiplexed Ion Beam Imaging (MIBI)[27–29] and Imaging Mass Cytometry (IMC)[30]. These platforms, which do not suffer from sample autofluorescence during imaging, are well-suited to reveal relationships between distinct cell phenotypes, their functional states, and overall tissue architecture[15,31–39], but limited in their ability to discern subcellular features. This is due to a limited imaging resolution, which is directly related to the spot size of the ion beam or laser (approximately 400 nm for MIBI and 1000 nm for IMC under standard conditions). We recently developed additional chemical probes for High Definition MIBI (HD-MIBI) for targeted MSI at even higher resolutions[24]. HD-MIBI can resolve targets at ~50 nm, but is currently limited to eight detectable parameters and difficult to scale across large areas of tissue.

We reasoned that a physically expanded sample compatible with the high-vacuum or desiccated natures of MIBI and IMC, respectively, would allow highly multiplexed imaging into the subcellular resolutions without costly instrumental development or new labeling chemistries. Given the abundance and accessibility of archival tissue samples, a desirable ExM-derived method should preserve protein epitopes to be compatible with high-plex antibody staining and imaging of these highly cross-linked samples. Previous ExM attempts in FFPE tissues required extensive proteolytic digestion after labeling to homogenize the tissue for uniform expansion[11]. This step is detrimental to preserving epitope antigenicity, as seen by lower antibody staining signals, which can impair high-plex protein-based staining and imaging[40]. An alternative method, known as Magnified Analysis of Proteome (MAP)[10], uses a high concentration of monomers to reduce the intermolecular crosslink between proteins and promote their linkage to the polymer network through non-enzymatic denaturation, but is not compatible with FFPE tissue yet. Preservation of antigenicity is fundamental for the successful integration of ExM-related hydrogel methods with high-plex immunohistochemistry.

We herein introduce Expand and comPRESS hydrOgels (ExPRESSO), a method to interrogate the proteome organization in physically expanded hydrogel-embedded tissues without proteolysis. ExPRESSO then facilities the removal of water molecules in the hydrogel while retaining the expanded lateral dimensions. We demonstrated the compatibility of these gels in the high-vacuum chamber of the MIBIscope (down to $10^{-7}$ mBar) and under the desiccated condition required for the IMC platform, obtaining antibody-based imaging of 23 proteins at nearly four times the original resolution without instrument modifications. Our optimized protocol retains tissue antigenicity and morphology for high-resolution antibody-based spatial proteomics in the subcellular domain of archival tissues. Application of ExPRESSO in FFPE lymphoid and brain tissues validated the conservation of biomolecular and cellular structures described previously, while resolving orchestrated features of multi-cellular organization and tissue architecture at the subcellular level, particularly that of the blood-brain barrier (BBB). ExPRESSO has the potential to be complementary to spatial-omic studies in addition to the standard light-based modalities, allowing a multi-dimensional understanding of how biomolecules, cells, and both subcellular and cellular neighborhoods are organized in health and disease.

## Results

### Development and evaluation of ExPRESSO hydrogels

To enable high-dimensional immunohistochemistry on archival tissue samples with MSI, it is necessary to develop an ExM approach compatible with the highly cross-linked nature of FFPE tissue sections which retains tissue structure and protein epitopes for high-plex antibody staining. In addition, the expanded hydrogels must be dehydrated while retaining their expanded nature to allow vacuum compatibility. Overcoming these challenges would further improve the resolution for high-plex spatial proteomics on platforms such as MIBI and IMC without the need for instrumental modifications.

In the previously reported Expansion Pathology (ExPath) framework to enable ExM on FFPE sections[11], the sections are first stained for target proteins in situ before the introduction of monomeric gel components, followed by gelation, proteolytic digestion, and gel expansion (Supplementary Fig. 1a, top). The ExPRESSO process entails initial sample gelation, a non-enzymatic denaturation, antibody staining, and gel expansion (Supplementary Fig. 1a, bottom). The enzymatic digestion steps used in the ExPath approaches can result in proteolytic fragmentation and dilution of labels. The avoidance of enzymatic digestion maintains (1) extracellular tissue structures, (2) protein epitope integrity, and (3) antibody signal intensity, maximizing cross-compatibility with previously validated antibodies on conventional FFPE samples.

Given the highly cross-linked nature of FFPE tissues, we incorporated a long heat-induced epitope retrieval (HIER) step to reverse these chemical crosslinks[41,42], followed by an optimized anchoring protocol for isometric expansion of these tissues without proteolytic digestion (Supplementary Fig. 1a, bottom). Various cancer tissue types retained tissue structure and epitope staining after this treatment as shown using a duplex immunohistochemistry staining against Vimentin and pan-Cytokeratin (Supplementary Fig. 1b).

A major challenge of using hydrogel-embedded tissue with MSI platforms, such as MIBI and IMC, is the prerequisite for dehydrated or vacuum-compatible samples to avoid ionic interference during imaging[43]. Since expanded gels are composed of at least 99% water, increasing the salt concentration[44] or typical dehydration processes will cause gel shrinkage or disruption of tissue morphology (Supplementary Fig. 2a). We reasoned that utilizing the negatively-charged nature of the polyacrylate ExM gel would allow strong adhesion onto positively-charged slides (a routine treatment to allow for tissue section adherence). This treatment, followed by gradual evacuation of water from the gel, would lead to a controlled compression in the Z-axis while maintaining the expanded X- and Y-axes. Such a compression approach is appealing because it fulfills two requisites, (1)

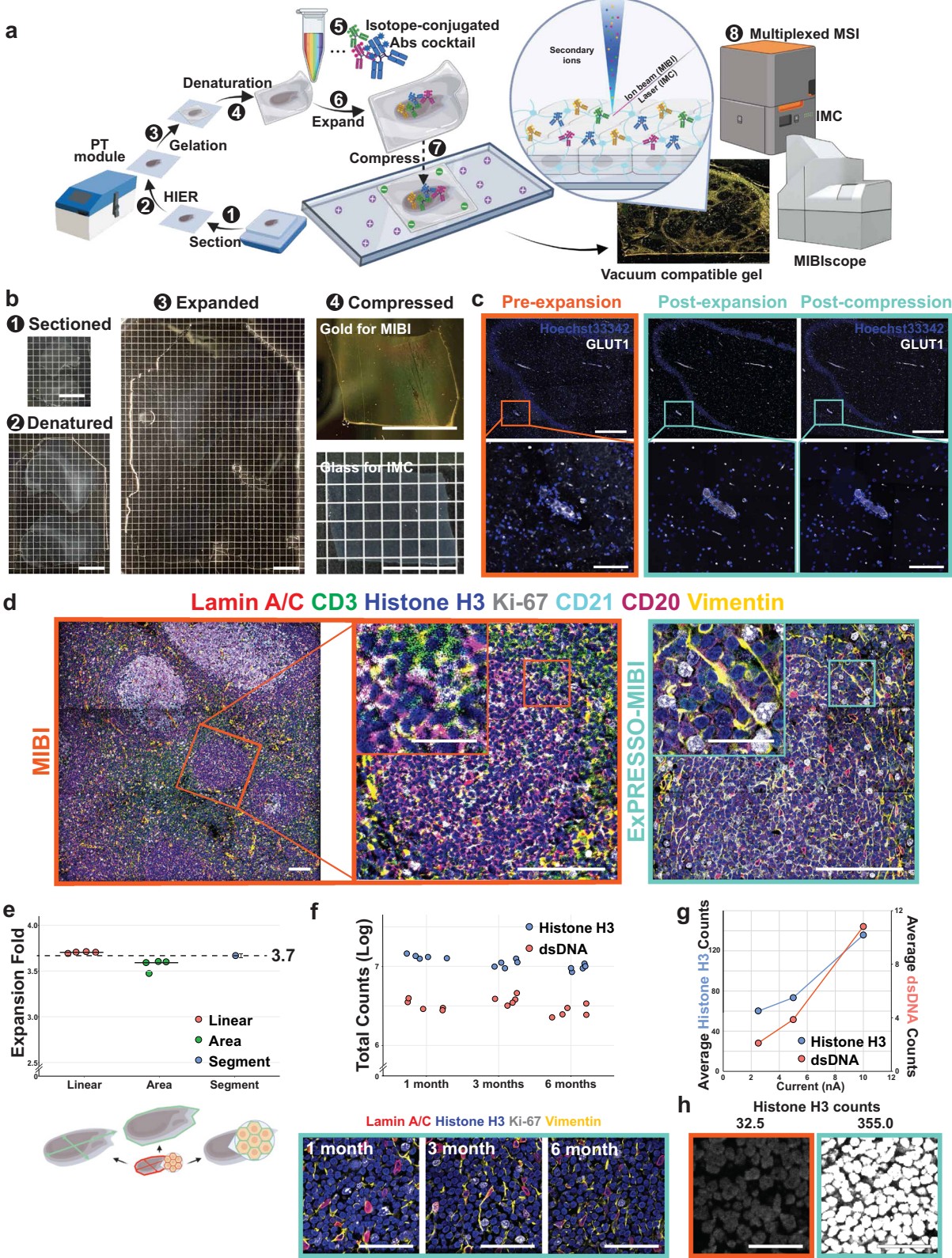

complete water removal and (2) retention of the intrinsic molecular organization at the nanometer scale in the Z-axis[45]. In ExPRESSO, we implemented this ExM protocol and subsequent compression approach (Fig. 1a, b). The compression of the fully expanded ExPRESSO hydrogels onto charged slides resulted in the retention of expanded X and Y dimensions, with no observable differences from the pre-compression gel (Fig. 1c).

Since water was completely evacuated from ExPRESSO gels, they retained their shape and presented a flat topology even under high-vacuum without detectable detachment or deformation (Supplementary Fig. 2b). Under the same imaging setting (identical beam current), secondary electron detector (SED) images were generated and visualized for non-ExPRESSO and ExPRESSO gels on the MIBIScope before and after MIBIScope data acquisition (Supplementary Fig. 2b),

**Fig. 1 | The ExPRESSO workflow for multiplex tissue imaging at 3.7× resolution.**
**a** The ExPRESSO workflow is compatible with (1) archival tissue sections, which are (2) reverse-cross-linked via heat-induced epitope retrieval, (3) embedded into a hydrogel, (4) denatured, (5) stained with isotope-labeled antibodies before (6) expansion and (7) compression in a desiccated environment. (8) ExPRESSO samples are compatible with the MIBIscope, IMC, and beyond. **b** Representative images of a human hippocampus section after (1) sectioning, (2) denaturing, (3) expanding, and (4) compressing onto gold and glass slides for MIBI (upper) and IMC (lower) analysis, respectively. Each grid is 2.5 mm. Scale bars indicate 10 mm in physical measurements. **c** Representative immunofluorescence images of a human hippocampus section stained for GLUT1 and nuclei in pre-expansion (left), post-expansion (middle), and post-ExPRESSO compression (right) states. Scale bars indicate pre-expansion dimensions based on a 3.7-fold expansion: 500 μm (top), 100 μm (bottom). **d** Comparison of MIBI (left and middle) and ExPRESSO-MIBI (right) human tonsil imaging in two consecutive sections. Representative MIBI images for CD3, CD20, CD21, Vimentin, Ki-67, Lamin A/C, and Histone H3. Scale bars indicate the pre-expansion dimensions based on a 3.7-fold expansion: 100 μm

(main), 25 μm (middle and right, enlarged). **e** Expansion fold quantification. Top: Independent quantifications from four tissues, represented with means and standard deviation. Bottom: Schematic representation of quantification strategies: (1) linear: ratio of the longest line in the tissue and its perpendicular, (2) area: ratio of the areas before and after ExPRESSO, (3) segmentation: ratio of segmented cell sizes between unexpanded and ExPRESSO tonsil tissues. **f** Long-term signal retention in ExPRESSO samples. The same ExPRESSO tonsil was imaged by MIBI after stored for 1, 3, and 6 months. Top: total histone H3 and dsDNA counts from 5 fields of view (FOVs) per time point. Bottom: representative MIBI images at each time. Scale bars indicate the pre-expansion dimensions based on a 3.7-fold expansion: 100 μm. **g** MIBI measurement of Histone H3 and dsDNA counts (per pixel) in an ExPRESSO tonsil analyzed with a range of primary ion currents. **h** Representative MIBI images of unexpanded and ExPRESSO-treated tonsils at equivalent spatial resolutions, capped at 300 counts per pixel. Values indicate counts (per pixel). Scale bars indicate the pre-expansion dimensions based on a 3.7-fold expansion): 25 μm.

---

demonstrating that ExPRESSO gels behave similarly to unexpanded tissue sections. Orthogonally, HeLa cells and FFPE tonsil sections were concurrently processed with both protein-retention ExM (proExM)/ExPath and our ExPRESSO protocols, before imaging on the MIBI-Scope. ExPRESSO linearly expands while preserving the overall structure and proteomic organization in FFPE tissue sections (Supplementary Fig. 2c), highlighting the superiority of ExPRESSO compared to conventional ExM protocols for multiplexed high-resolution histopathology using MSI approaches.

We further monitored potential sample distortion during the compression step of ExPRESSO as previously reported[10,46]. After compression, the distortion remained minimal compared to the pre- and post-expansion samples (Supplementary Fig. 2d, e, Supplementary Fig. 3a, b). We did observe distortion along the edges of the gel, even at the macroscopic level (Supplementary Fig. 2d), although these distortions subsided as we moved towards away from the edges (Supplementary Fig. 2d), and it is possible to overcome this with better gel handling and buffered area around tissue in the future. While the distortion comparison between pre-expansion to post-compression shows a buildup of distortions along with all processing steps, the overall distortion is limited (Supplementary Fig. 2e and Supplementary Fig. 3b). We next compared conventional MIBI and ExPRESSO-MIBI imaging on adjacent tonsil sections under identical instrument conditions. We observed that ExPRESSO-MIBI imaging resulted in a notable improvement in spatial resolution with no loss of architecture of a B cell follicle (Fig. 1d). The beam resolution was similar for both conditions, thus the increase in sample resolution was directly attributed to the sample expansion fold (Supplementary Fig. 3c). We further exemplified the additional resolving power of ExPRESSO-MIBI over conventional MIBI by resolving the association between Cajal and PML bodies, two nuclear bodies that usually associate with each other (Supplementary Fig. 3d). While the increased resolution benefit is apparent, a potential downside is the increased acquisition time (a function of the square of the expansion factor) to acquire the same total tissue area when compared to a conventional, non-expanded sample. Measurements with a profilometer indicated that the compressed ExPRESSO gel was approximately 780 nm thick (Supplementary Fig. 4a). Opposed to conventional, non-expanded tissue sections under vacuum, ExPRESSO gels exhibit a smooth topography and $^{12}C$ counts during multi-layer runs (Supplementary Fig. 4b).

We next implemented three strategies to determine the extent of lateral expansion in ExPRESSO methodology (Fig. 1e). Linear dimension measurements of the pre- and post-ExPRESSO tissues, area calculations of tissue space occupied pre- and post-ExPRESSO and cell size distribution based on cell segmentation of original and post-ExPRESSO tonsil images were concordant and indicate that the lateral expansion of ExPRESSO is approximately 3.7-fold (Fig. 1e).

A benefit of the lanthanide tags used for MIBI and IMC is that these elemental probes are stable for long periods of time. When stored appropriately in a vacuum desiccator, ExPRESSO tissues stained with lanthanide-tagged antibodies were re-imageable even after 6 months, with relatively consistent signal (Fig. 1f). These results highlight the versatility and potential of ExPRESSO in applications related to spatial analysis of the cellular and biomolecular composition of tissue samples.

There was a proportional relationship between ion dose from the primary beam and secondary ions detected on ExPRESSO gels analyzed by MIBI (Fig. 1g), in line with previous observations in MIBI samples[29]. We thus explored the possibility of increasing the beam current by fourfold to 10 nA for ExPRESSO sample acquisition, and achieved a comparable resolution as the one obtained in unexpanded samples imaged at 2.5 nA (Supplementary Fig. 4c). Matching imaging resolution in this manner effectively resulted in more than fourfold increase in detected ion counts on ExPRESSO samples, with the same imaging time and effective resolution (Fig. 1h). These data indicate the flexibility for ExPRESSO to obtain higher signal counts at the same resolutions and acquisition rates used in a typical MIBI experiment. Conversely, imaging speed can be substantially accelerated during ExPRESSO imaging to achieve resolutions and signal counts comparable to unexpanded samples (Supplementary Note 1). Together, these results demonstrate improvements in MSI-based spatial proteomic imaging resolutions, speed, and signal yield enabled by ExPRESSO gels.

## ExPRESSO advances mass spectrometry imaging resolution to enable detailed interrogation of archival tissue samples

We leveraged the multiplexed capabilities of the MIBI to evaluate epitope retention of tissues between conventional staining and ExPRESSO-treatment on adjacent human tonsil sections. Samples were stained with 24 key lineage-specific and tissue structural markers (Fig. 2a, Supplementary Figs. 5 and 6). There was consistent staining and tissue distribution patterns between these conditions, and notable improved resolution in the ExPRESSO-treated sections. We note the T cell markers CD3, CD4, and CD8 show clear non-overlapping CD4 and CD8 expression (Fig. 2a), while CD20, Ki-67, and CD21 were appropriately found within the B-cell follicles (Fig. 2a). Additionally observed were the distinctive CD68 expression on macrophages, structural markers Vimentin and Lamin A/C, and the enrichment of HLA-DR and Pax-5 within B-cell follicles (Fig. 2a). Similar improvements of applying ExPRESSO on the IMC platform were seen (Fig. 2b and Supplementary Fig. 7a), demonstrating the consistency and versatility of our method.

Line-scan quantification of unexpanded and ExPRESSO-treated sections in MIBI and IMC images confirmed that ExPRESSO resulted in enhanced ability to distinguish between cell and tissue structures, such as Lamin A/C, Vimentin, CD21, and Histone H3 (Fig. 3a and Supplementary Fig. 8a). There was an unexpected improvement in the signal-

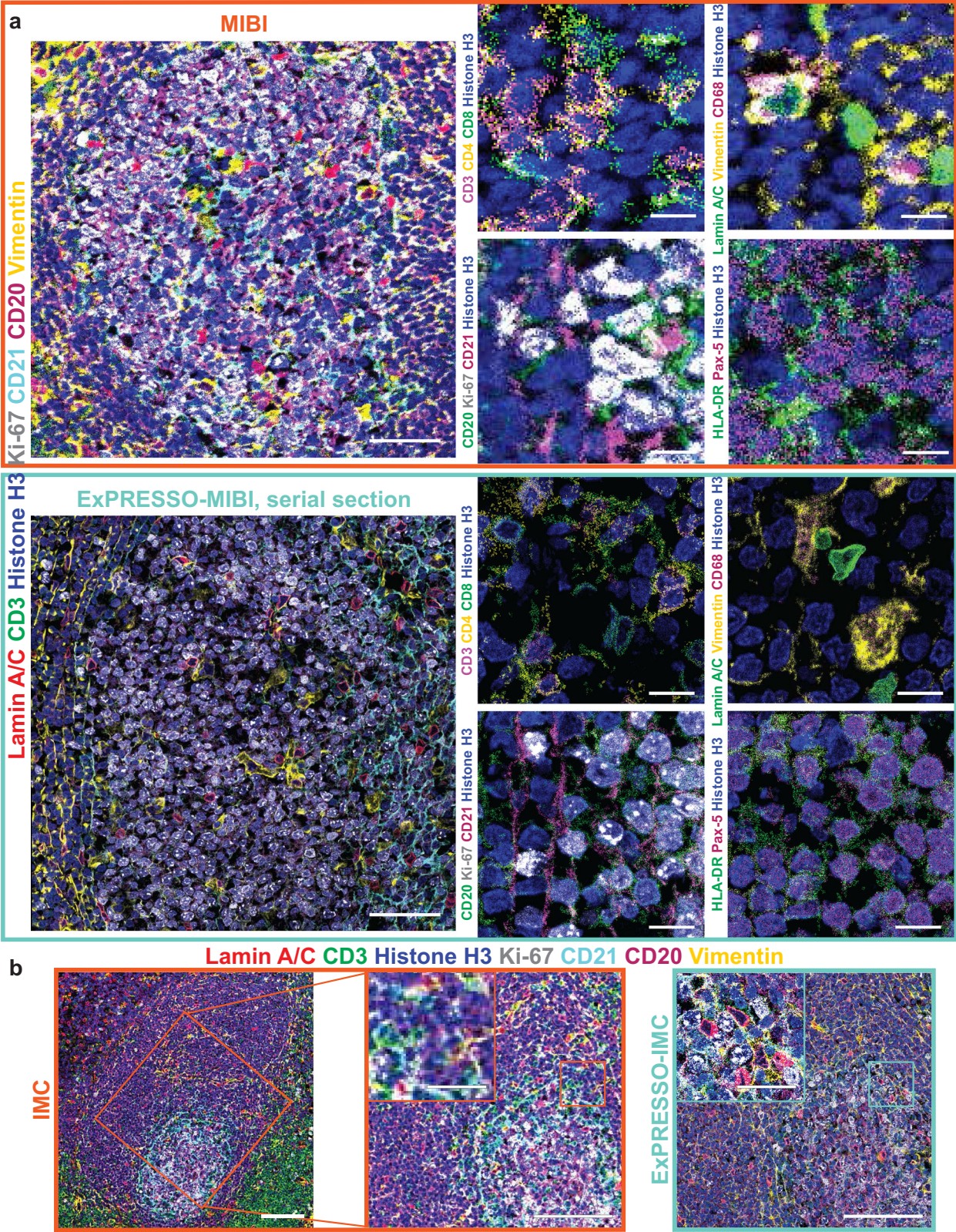

to-noise ratio, likely due to the clearance of tissue lipids and extracellular matrix during the denaturation process (Fig. 3a and Supplementary Fig. 8a). The increased resolution revealed certain subcellular structures that were previously challenging to image using MIBI and IMC, including detailed plasma membrane structures, mitochondrial networks, the nucleolus, and nuclear clusters of H3K9ac enrichment (Fig. 3b, c and Supplementary Fig. 8b, c). ExPRESSO also improved cell

segmentation performance, as exemplified using a neural network segmentation method, MESMER, for cell feature extraction and quantification[47] (Fig. 3d and Supplementary Fig. 8d), and retained the capabilities for cell phenotyping (Fig. 3e and Supplementary Fig. 8e). Taken together, these experiments demonstrate that ExPRESSO enables high-plex protein profiling in archival tissue samples at enhanced resolutions.

**Fig. 2 | ExPRESSO is compatible with mass spectrometry-based highly-multiplex imaging technologies. a** Comparison of MIBI (top) and ExPRESSO-MIBI (bottom) imaging of two sections of a human tonsil with markers for helper T cells (CD3 and CD4), cytotoxic T cells (CD3 and CD8), B cells (CD20 and Pax-5), B cells and follicular dendritic cells (CD21), antigen-presenting cells (HLA-DR), macrophages (CD68), structural intermediate filaments (Vimentin), proliferation (Ki-67), nuclear envelope (Lamin A/C), and nuclei (Histone H3). The four images on the right are enlarged views of the image on the left. Scale bars (the ExPRESSO-MIBI scale bars indicate the pre-expansion dimensions based on a 3.7-fold expansion): 50 μm (left), 10 μm (right). **b** Comparison of IMC (left and middle) and ExPRESSO-IMC (right) imaging in two adjacent sections of a human tonsil with markers for T cells (CD3), B cells (CD20), B cells and follicular dendritic cells (CD21), structural intermediate filaments (Vimentin), proliferation (Ki-67), nuclear envelope (Lamin A/C), and nuclei (Histone H3). The middle image is an enlarged view of the image on the left, and its size was matched with the right image. Enlarged views are shown at the top left in the middle and right images. Scale bars (the ExPRESSO-IMC scale bars indicate the pre-expansion dimensions based on a 3.7-fold expansion): 100 μm (main), 25 μm (middle and right, enlarged).

## ExPRESSO resolves brain tissue features

We next performed ExPRESSO-MIBI imaging using a panel of 21 markers on archival human brain tissues originating from the middle frontal gyrus and hippocampus. The staining patterns of these antibodies were consistent with the Human Protein Atlas[48] and our recent MIBI brain imaging study[38]. We observed comparable staining patterns between unexpanded and ExPRESSO-treated sections of brain tissues (Supplementary Fig. 9a). This further confirmed the retention of protein epitopes in our proteolytic digestion-free method and the applicability of ExPRESSO across various tissue types. Notable, typical star-shaped morphology of astrocytes is revealed with Glutamine Synthetase (GlnSyn) and Glial Fibrillary Acidic Protein (GFAP) markers; distinct microglial shapes are observed with Iba1 staining; cortical neurons of the middle frontal gyrus and granule cells of the hippocampus are seen through markers such as MAP2 or CD56; and vasculature is demarcated by glucose transporter 1 (GLUT1), CD105 (Endoglin) and smooth muscle actin (SMA) (Fig. 4a, b, Supplementary Fig. 9a).

We next used MIBI to image a strip of ExPRESSO-treated middle frontal gyrus tissue, traversing from the gray to white matter (Fig. 4a and Supplementary Fig. 10a). Visual and quantitative analysis of markers that are differentially expressed between the gray and white matter confirmed their expected patterns (Fig. 4a, Supplementary Fig. 10a, b; gray matter: GFAP and GlnSyn high; white matter: GFAP and GlnSyn low). We further observed an improvement in resolving cells and features around the BBB (Fig. 4b). Landmark cell types and features in the brain were identified, including microglia (Iba1⁺), vasculature (GLUT1⁺), astrocytes (GlnSyn⁺) and the contacts between astrocytic end-feet and endothelial cells around the BBB (GlnSyn⁺ and GLUT1⁺, respectively) (Fig. 4c). ExPRESSO-IMC also demonstrated similar capabilities to identify and finely resolve the brain cytoarchitecture (Supplementary Fig. 10c).

Although deep learning-based nuclear segmentation is generally used for high-dimensional tissue imaging cell type analysis[14,15,28,34,39,47] (Supplementary Fig. 11a), this approach is not suitable for the identification of larger brain structures and features. We harnessed a different approach to identify tissue-level features at scale (Supplementary Fig. 11b, and Methods), thus incorporating both single-cell data and non-nuclear feature-level information (Supplementary Fig. 11b). Using this two-pronged approach, we were able to distinguish intricate brain structures while retaining cellular identity, including separating out epithelial, microglial, neuronal cells and astrocytes from blood vessel features, microglia features and astrocyte features (Fig. 5a).

## ExPRESSO unravels the blood-brain barrier organization

We next focused on resolving the complex organization of the BBB. The BBB is a barrier of endothelial cells that selectively allows the passage of ions, macromolecules, and cells. We postulated that an agglomerated analysis of all ExPRESSO-MIBI BBB structures with our single-cell and non-nuclear feature-level combinatorial analytical approach would allow a reconstruction of the molecular organization at the BBB interface. This computational approach simulates a "walk" from identified endothelial features "outward" away from the vasculature or "inward" toward the lumen of the vessels (Supplementary Fig. 11c).

In unexpanded MIBI multiplexed images of BBB structures in a brain sample from a patient diagnosed with Alzheimer's disease (AD) ($n = 281$), we were unable to computationally (Fig. 5b, left) or visually discern BBB organization (Supplementary Fig. 12a). Analysis of ExPRESSO-MIBI resolved BBB structures ($n = 103$) across 50 field of views (FOVs) revealed a clear, structured ordering of proteins involved in BBB organization (Fig. 5b, right). We observed the presence of Serum Amyloid A (SAA), Fibrinogen, and Albumin originating near the lumen of the BBB toward the brain (Fig. 5b, right). These factors generally do not traverse beyond the BBB from within the vessels in healthy brain[49–51], thus suggestive of a loss in BBB barrier integrity.

We also observed the von Willebrand Factor (vWF), a glycoprotein synthesized within endothelial cells and a key regulator of hemostasis[52] (Fig. 5b, right). A tight enrichment of GFAP, CD105 and GLUT1 was observed beyond the region enriched in vWF (Fig. 5b, right). CD105 and GLUT1 can frequently be found on the surface of blood vessels[53,54], whereas GFAP, a filamentous protein classically used to identify astrocytes, recapitulates the localization of astrocyte end-feet between neurons and blood vessels to mediate neurovascular coupling and signal relay[55] (Fig. 5b, right). Type IV collagen is often found in the vascular basement membrane, which separates the endothelial cells from neurons; it was appropriately detected in this region (Fig. 5b, right). The proximity of astrocytes to the BBB was also demonstrated by the presence of GlnSyn (Fig. 5b, right), a key protective enzyme enriched in astrocytes to protect it against neuronal excitotoxicity[56]. Lastly, we detected Iba1 localization reflective of the presence of microglia near BBBs (Fig. 5b, right). This is a key indicator of their important perivascular localization in sampling the influx of blood-borne agents into the central nervous system via the BBB[57] (Fig. 5b, right). This analysis demonstrates that ExPRESSO-MIBI and computational approaches can be used to resolve the organization of the BBB within the native tissue context.

We further applied an expanded ExPRESSO panel to include Amyloid beta (Aβ) and tau markers to a larger cohort of AD and non-AD patient samples (Supplementary Table 1). We observed clear Aβ plaques and tau neurofibrillary tangles discrepancies in brain samples between patients with low Braak scores (non-AD; Fig. 6a top) and high Braak scores (AD; Fig. 6a bottom and 6b). We were able to confirm the staining consistency of BBB features (Fig. 6b), and observed Aβ plaques and Tau tangles within the native tissue context (Fig. 6a, b, inserts) at resolutions beyond that of conventional MIBI imaging. Application of our anchoring methodology and feature extraction pipeline developed above on the BBB of non-AD and AD brain samples allowed the dissection and comparison of spatial features around the BBB (Supplementary Fig. 12b). These results further highlight the framework for spatial dissection of subcellular features within the native tissue and disease context.

## Discussion

ExPRESSO is a method that enables expansion microscopy for analytical imaging technologies previously unable to accommodate hydrated samples, such as MSI. The ExPRESSO framework first expands

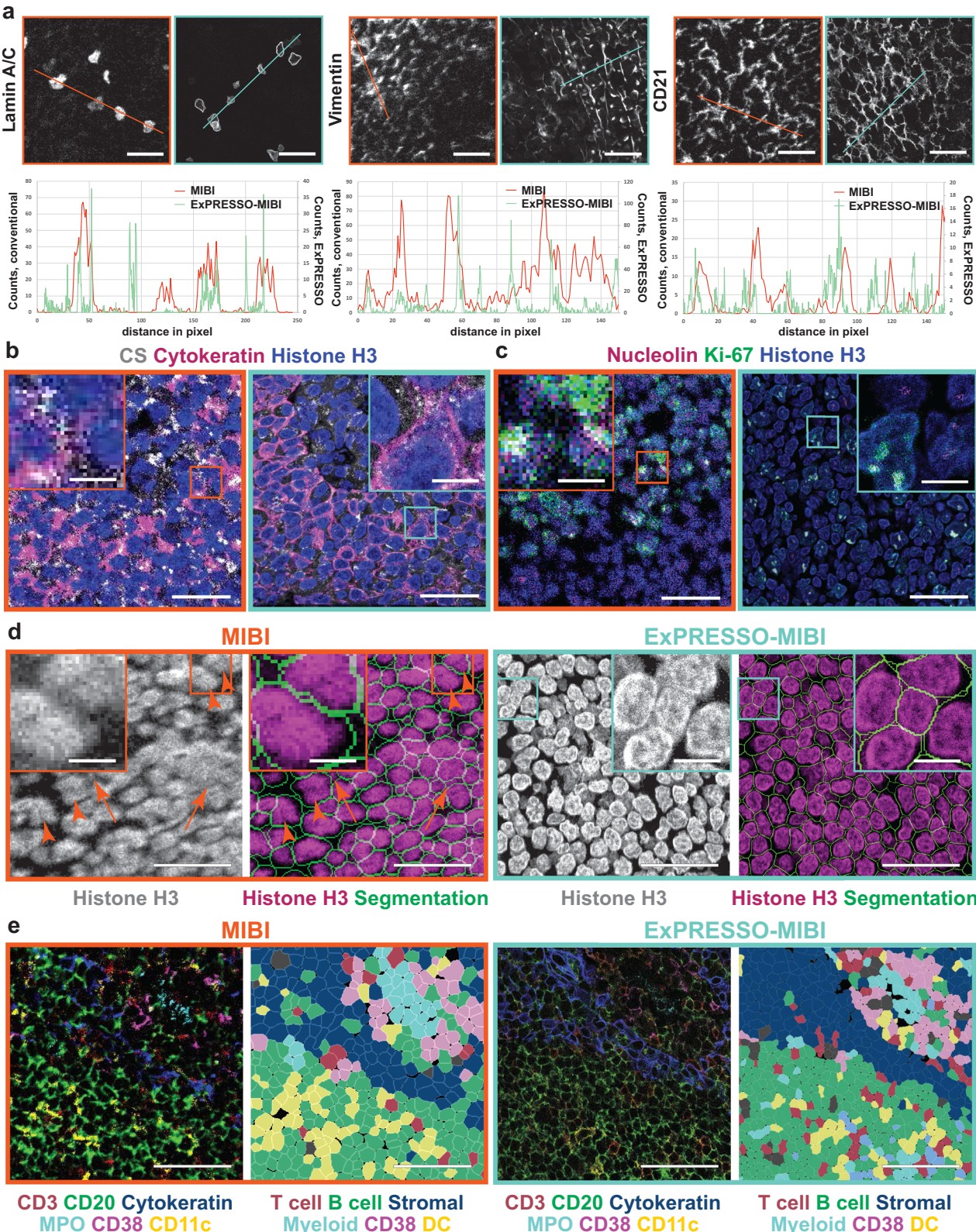

tissue sections by ~3.7-fold, followed by antibody cocktail staining, and a final controlled hydrogel compression in the Z-axis for complete water removal. This allows for previously unattainable spatial resolutions and high signal-to-noise sample analysis, using commercially available mass spectrometry-based high-plex tissue imaging instruments without cost-prohibitive and challenging engineering modifications. The compatibility of our method with archival FFPE tissues while maintaining antigenicity will be a key step towards leveraging large clinical cohorts. We applied ExPRESSO and a panel of 24 markers to human tonsils, reproducing tissue staining patterns performed in parallel on unexpanded samples with approximately 3.7-fold magnification, revealing cellular features previously unresolvable by MIBI and IMC. We next applied ExPRESSO to brain samples, a notoriously challenging tissue to image due to high tissue autofluorescence[58] and

**Fig. 3 | ExPRESSO better resolves fine details of subcellular structures.**
**a** Representative MIBI images for Lamin A/C (left), Vimentin (middle), and CD21 (left) in human tonsils. Line scans along the orange (MIBI) and cyan (ExPRESSO-MIBI) lines in the MIBI images. Raw ion counts (Y axis) were plotted to each individual pixel (X axis), and the ExPRESSO-MIBI lines were scaled down by expansion fold to match MIBI lines. Scale bars (the ExPRESSO-MIBI scale bars indicate the pre-expansion dimensions based on a 3.7-fold expansion): 25 μm. **b, c** Comparison of MIBI (left) and ExPRESSO-MIBI (right) imaging in two adjacent sections of a human tonsil. ExPRESSO-MIBI resolves **b**) fine structure of cytokeratin, citrate synthase (CS) puncta, and **c** Ki-67 and nucleolin compartmentalization. Scale bars (the ExPRESSO-MIBI scale bars indicate the pre-expansion dimensions based on a 3.7-fold expansion): 25 μm (main), 5 μm (enlarged). **d** Comparison of cell segmentation performance in MIBI (left) and ExPRESSO-MIBI (right) imaging of a human tonsil.

For each group, representative MIBI images on the left show the nuclei by Histone H3 staining, and images on the right depict a cell segmentation map together with nuclear staining (Histone H3). Orange arrows indicate regions with high nuclear signal that were not segmented as cells, and arrowheads point to doublets indicative of undersegmentation. Scale bars (the ExPRESSO-MIBI scale bars indicate the pre-expansion dimensions based on a 3.7-fold expansion): 25 μm (main), 5 μm (enlarged). **e** Comparison of cell phenotyping in MIBI (left) and ExPRESSO-MIBI (right) imaging of a human tonsil. For each group, representative MIBI images on the left show CD3 (red), CD20 (green), Cytokeratin (blue), MPO (cyan), CD38 (magenta), and CD11c (yellow). Images on the right depict the cell phenotype map with major cell types, which were clustered after segmenting single cells. Scale bars (the ExPRESSO-MIBI scale bars indicate the pre-expansion dimensions based on a 3.7-fold expansion): 50 μm.

intricate cell and tissue morphologies. We demonstrated that ExPRESSO could resolve cellular features in human brain samples, such as Aβ plaques and Tau tangles in AD-diseased brains and the characteristic morphologies of astrocytes, neurons, microglia and blood vessels. This allowed for previously unattainable increases in resolution and measurable features that revealed intricate structures within the BBB. Such quantitative imaging and assessments at high-dimensions and resolutions in situ within biological samples will continue to advance our understanding of tissue biology.

A key limitation of most expansion microscopy-derived methods is the necessary protease digestion step[4,5,11], which can be detrimental to many epitope-targets of antibodies used for multiplexed immunohistochemistry. The ExPRESSO workflow does not include a proteolytic treatment. Instead, we incorporated a prolonged reverse crosslinking step, optimized anchoring step, and a detergent-based denaturation step to maximize epitope retention while retaining isotropic hydrogel expansion. This anchoring approach was largely inspired by the recent MAP methodology[10,59], although MAP was not compatible with FFPE tissues in our hands. Our initial ExPRESSO screen showed a 80% success rate (58 antibodies out of 73) using FFPE antibody clones routinely used in highly-multiplex tissue imaging studies by our laboratories and others[15,24,28,31,34,38,39,60,61]. Although this study focused on the proteome, interrogation of other molecular species, including nucleic acids[34,62], lipids[63], glycans[26], drugs[24], and other small molecules[64] should be possible in ExPRESSO-processed tissue samples.

Another limitation of expansion microscopy-derived methods, including ExPRESSO, is the reduction of detectable labels per pixel unit within a fixed imaging field of view, which can be further aggravated by the protease digestion step often used in expansion-related protocols[5,11]. Despite the theoretical signal dilution, quantification of all markers was higher than expected in the ExPRESSO tonsil (Supplementary Table 10) and in particular 8 out of 17 antibodies had a higher count per pixel over the same non-expanded area, suggestive of additional factors in play such as increased epitope accessibility by molecular decrowding[65]. Still, we observed that certain membrane proteins (e.g., CD20, CD11b, CD45) had lower counts per pixel compared to the non-expanded samples (although the overall counts are higher than expected considering dilution by expansion), which highlights the need for extensive titration and careful validation of antibody panels for ExPRESSO. In SIMS, signal intensity is not only depending on the enrichment of the readout element but also affected by the local composition of the sample. Still, protein level quantification is robust by MIBI[29], and the multiparameter and high-resolution capabilities of ExPRESSO enable harnessing relative expression and spatial location for quantitative and qualitative description of the samples of interest.

The improved spatial resolution obtained via ExPRESSO is primarily dependent upon the selected hydrogel strategy and instrument settings. In this study, we used a monomer solution of acrylamide and sodium acrylate at a ratio that limits magnification to about 3.7-fold. Modifications to this ratio by monomer chemistry[7–9] and the

application of iterative approaches using cleavable monomers[6] can lead to greater lateral resolutions. Improvements in resolution might also be attained with refinements and developments in instrument capabilities. In the experiments reported in this manuscript, we leveraged oxygen and xenon ion sources for tissue imaging on the MIBI at a resolution of about 400 nm[29] and a laser with a spot size of approximately 1000 nm on the IMC[30]. We have previously demonstrated protein imaging at a resolution down to 50 nm in single cells using a cesium ion source[24]. Similarly, a radio frequency plasma oxygen primary ion source has also achieved resolutions down to 40 nm[66]. Such instrumental capabilities, combined with ExPRESSO, will enable antibody barcoding for single-molecule detection and interrogation of an exponentially larger number of targets. ExPRESSO also holds promise for improved resolution of large scale proteomics methods relying on microdissection[67,68] and liquid microjunction[69,70]. Although the compressing step allows the ExPRESSO gels to withstand high-vacuum environments with an apparent sacrifice of axial resolution, MIBI and many other analytical methods have superior axial resolution over lateral resolutions with the 5–100 nm range[24,29], allowing it to recapture axial resolutions, opening the way to 3D depth profiling in thick tissue sections, although this would require extensive further validation experiments. Recently, an approach for IMC 3D tissue imaging reconstruction was performed by co-registering hundreds of stained and imaged serial tissue sections[71]; a similar approach may resolve the 3D tissue architecture with ExPRESSO-MIBI or ExPRESS-IMC with significant investment of instrument time and resources.

In this manuscript, we have demonstrated that ExPRESSO is compatible with MIBI and IMC. Both platforms use the same isotopes for detection of antibodies, albeit with unique sensitivities for different metal elements[29,72]. Thus, cross-compatibility between the two platforms is possible but requires additional validation and titration of the antibody panel. Compared with non-expanded samples, while the acquisition time increases exponentially with the physical expansion factor albeit with some alternative workarounds (Supplementary Note 1), the gel-embedded nature of the gel could be a potential area of interest for future work in minimizing the distortion observed in traditional SIMS analysis (Supplementary Fig. 3d)[73,74]. Specifically, adherent cells are challenging to image using the conventional MIBIscope, due to uneven etching of the cells and its surrounding conductive gold slide, and often leads to decreased beam penetration and signal yield. In contrast, the uniform nature of specimen-embedded ExPRESSO gels allows more uniform ion beam etching for a more homogeneous imaging process (Supplementary Fig. 4b).

In summary, ExPRESSO enables analyses of expanded samples with mass spectroscopy-based methods in spatial resolutions and dimensions previously unattainable due to physical limitations, thus allowing the interrogation of cells and features at a subcellular resolution while maintaining their native tissue context. ExPRESSO is applicable to a range of biological samples and can potentially reveal fundamental principles of cellular and tissue organization in both health and disease from the molecular scale on up.

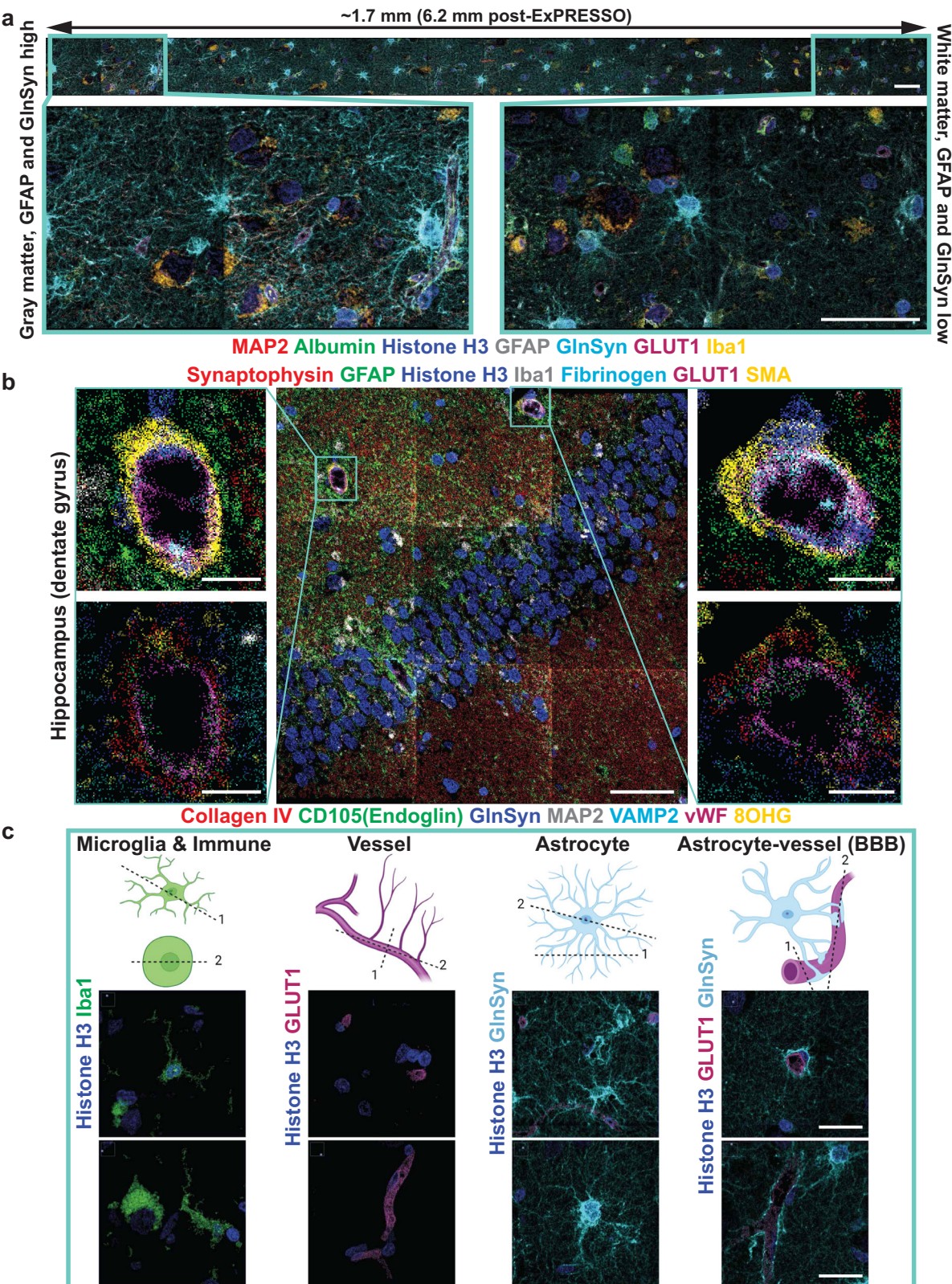

## Methods

### Ethical statement

Archival human FFPE tissues, including tonsil, tumor (of spleen, kidney, intestine, brain, and pancreas), and brain (hippocampus and middle frontal gyrus were from the same donor) were obtained from Stanford Pathology Department. Specifically, for the brain sections from Stanford Pathology, the postmortem interval of the brain tissue was 2.8 hours. Clinical diagnosis and neuropathological scores were generated by Stanford clinicians and pathologists. The clinical diagnosis was based on DSM-IV criteria. AD neuropathologic change and severity scores were evaluated by NIA-AA guidelines[75,76]. Neuropsychological test results within 1 year of death were in the upper three quartiles for the study. For the paired non-AD and AD patients middle frontal gyrus and hippocampus sections, archival human

**Fig. 4 | ExPRESSO enables multiplexed proteomic assessment of archival human brain sections at subcellular resolution. a** MIBI imaging of an ExPRESSO-processed human brain tissue section from the middle frontal gyrus. Top: representative linear acquisition of 16 FOVs, each one with a size of 400 × 400 μm (physical measurements). Images include markers for neurons (MAP2), vessel lumen and BBB leakiness (Albumin), astrocytes (GFAP and GlnSyn), vessels (GLUT1), microglia (Iba1), and nuclei (Histone H3). Bottom: enlarged views of the gray matter (left; GFAP and GlnSyn high) and white matter (right; GFAP and GlnSyn low). Scale bars (the ExPRESSO-MIBI scale bars indicate the pre-expansion dimensions based on a 3.7-fold expansion): 50 μm. **b** MIBI imaging of an ExPRESSO-processed human hippocampus tissue section. Middle: Representative tile acquisition of nine FOVs, each one with a size of 400 × 400 μm (physical measurements). Images include markers for pre-synapse (Synaptophysin), astrocyte projections (GFAP), nuclei (Histone H3), microglia (Iba1), vessel lumen, and BBB leakiness (Fibrinogen), vessels (GLUT1), muscularized vasculature (SMA). Left and right: Enlarged views of the blood-brain barrier (BBB). The BBB images at the top include the markers in the middle image and the BBB images at the bottom include markers for basement membrane (Collagen IV), endothelial cells (CD105(Endoglin), and vWF), astrocytes (GFAP), neurons (MAP2), pre-synapse (VAMP2), and neurons with oxidative stress (8OHG). Scale bars (the ExPRESSO-MIBI scale bars indicate the pre-expansion dimensions based on a 3.7-fold expansion): 50 μm (middle) and 10 μm (left and right). **c** Representative MIBI images of cellular and non-nuclear components in the human brain. From left to right, images highlight microglia, vessels, astrocytes, and BBB. Vessels are observed as transversal (top) and longitudinal (bottom) sections. Scale bars (the ExPRESSO-MIBI scale bars indicate the pre-expansion dimensions based on a 3.7-fold expansion): 25 μm.

postmortem FFPE tissue was obtained deidentified from Brigham and Women's Hospital (BWH). Tissue was from 6 individuals, 3 carrying a clinical diagnosis of Alzheimer's disease (AD) and 3 without clinical AD, balanced by sex and age (Supplementary Table 1), however, sex and gender were not considered in the study design due to the proof-of-concept nature of this methodological study. The tissue represented a range of AD pathology Braak stages by evaluation at BWH, based on NIA-AA guidelines[75].

The tonsil and tumor samples from Stanford Pathology were sectioned at 2 μm thickness, the brain samples were sectioned at a thickness of 4 μm, while samples from BWH has a thickness of 5 μm.

## Antibody conjugation and panel

Antibody conjugation was performed as previously described[77] using the Maxpar X8 Multimetal Labeling Kit (Fluidigm, 201300) and Ionpath Conjugation Kits (Ionpath, 600XXX) with slight modifications to manufacturers' protocols. In short, 100 μg BSA-free antibody was first washed with the conjugation buffer, then reduced using a final concentration of 4 μm TCEP (Thermo Fisher Scientific, 77720) to reduce the thiol groups for 30 min in a 37 °C water bath. The reduced antibody was then mixed and incubated with lanthanide-loaded polymers for 90 min in a 37 °C water bath, then washed five times with an Amicon Ultra filter (Millipore Sigma, UFC505096). The resulting conjugated antibody was quantified using a NanoDrop (Thermo Scientific, ND-2000) in IgG mode, at 280 nm, and the final concentration was adjusted to at least 30% v/v Candor Antibody Stabilizer (Thermo Fisher Scientific, NC0414486). Samples were stored at 4 °C. Details of the antibody panels are in Supplementary Tables 2–6.

## Gold slide preparation

The protocol of preparing gold slides has been described previously[28,29,60]. In short, Superfrost Plus glass slides (Thermo Fisher Scientific, 12-550-15) were first soaked and briefly supersonicated in dish detergent diluted in doubly distilled water (ddH2O), cleaned using Microfiber Cleaning Cloths (Care Touch, BD11945) then rinsed in ddH2O to remove any remaining detergent. After that, the slides were air-dried with a constant stream of air in the fume hood. The coating of 30 nm of tantalum followed by 100 nm of gold was performed by the Microfab Shop of Stanford Nano Shared Facility (SNSF) and New Wave Thin Films (Newark, CA).

## Vectabonding

To introduce positive charges for better adhesion of gels or tissue sections onto the surface, pre-cleaned glass slides or the gold slides were silanized with Vectabond Reagent (Vector Labs, SP-1800-7) as per the protocol supplied by the manufacturer. The slides were first soaked in neat acetone for 5 min, then transferred into 1:50 diluted Vectabond Reagent in acetone and incubated for 10 min. After that, slides were quickly dipped in ddH2O to quench and remove remaining

reagents, then tapped on a Kimwipe to remove remaining water. The resulting slides were air-dried then stored at room temperature.

## ExPRESSO protocol

All chemicals used were purchased from Sigma-Aldrich and used without further purification, if not specified.

**Antigen retrieval and hydrogel embedding.** FFPE tissue blocks were sectioned onto glass slides at the Stanford Pathology Core. Slides with FFPE sections were first baked in a dry oven (VWR, 10055-006) for 1 h at 70 °C, then were transferred into neat xylene and incubated for 10 min followed by transfer into xylene and incubation for another 10 min. Standard deparaffinization was performed with a linear stainer (Leica Biosystems, ST4020) in the following sequence: three times in xylene, three times in 100% EtOH, twice in 95% EtOH, once in 80% EtOH, once in 70% EtOH, and three times in ddH2O, 180 s each dip. Antigen retrieval was then performed at 97 °C for 40 mins with Target Retrieval Solution (Agilent, S236784-2) on a PT Module (Thermo Fisher Scientific, A80400012).

After removal from the PT Module, the cassette with slides and solution was left on the benchtop until it reached room temperature. Slides were rinsed with 1× PBS, then soaked in 30% (w/w) acrylamide in 1× PBS at 37 °C for 18 h. Tissue sections were then washed with 1× PBS for 5 min.

Monomer solution (1 × PBS, 2 M NaCl, 8.6% (w/w) sodium acrylate, 2.5% (w/w) acrylamide and 0.10% (w/w) N,N'-methylenebisacrylamide (BIS)), Ammonium persulfate (APS), and N,N,N',N'-Tetramethyl ethylenediamine (TEMED) solutions were prepared as previously described[5]. Tissue sections were then incubated with monomer solution for 10 min at room temperature. Meanwhile, a gelation chamber for thin gel was prepared with two strips of #0 coverslip (EMS, 72198-10) as spacers along with one piece of glass slide, similarly to the 10× ExM protocol[8].

The monomer solution was aspirated from the tissue sections carefully, then the slide with the sections was flipped and bridged between the two strips of #0 coverslip to form the gelation chamber. TEMED then APS was mixed with a new tube of monomer solution on ice, then around 30–50 μL of this solution was added through the gap between glass slide and slide with sections to avoid formation of bubbles. This gelation chamber was then incubated in a humidity chamber at 37 °C for 1 h.

**Gel denaturation and cleaning.** Gelation chamber was carefully disassembled with a clean razor blade, then slides with gel-embedded tissue sections were transferred into denaturing buffer (200 mM SDS, 200 mM NaCl, and 50 mM Tris in ddH2O water, pH 9.0) and incubated for 18 hours at 70 °C in a water bath. Denaturing buffer was renewed, and the gel was incubated with the denaturing buffer further at 95 °C for 1 h in a PT Module. After the solution was cooled to room temperature, the denaturing buffer was then replaced with 1× PBS with 1%

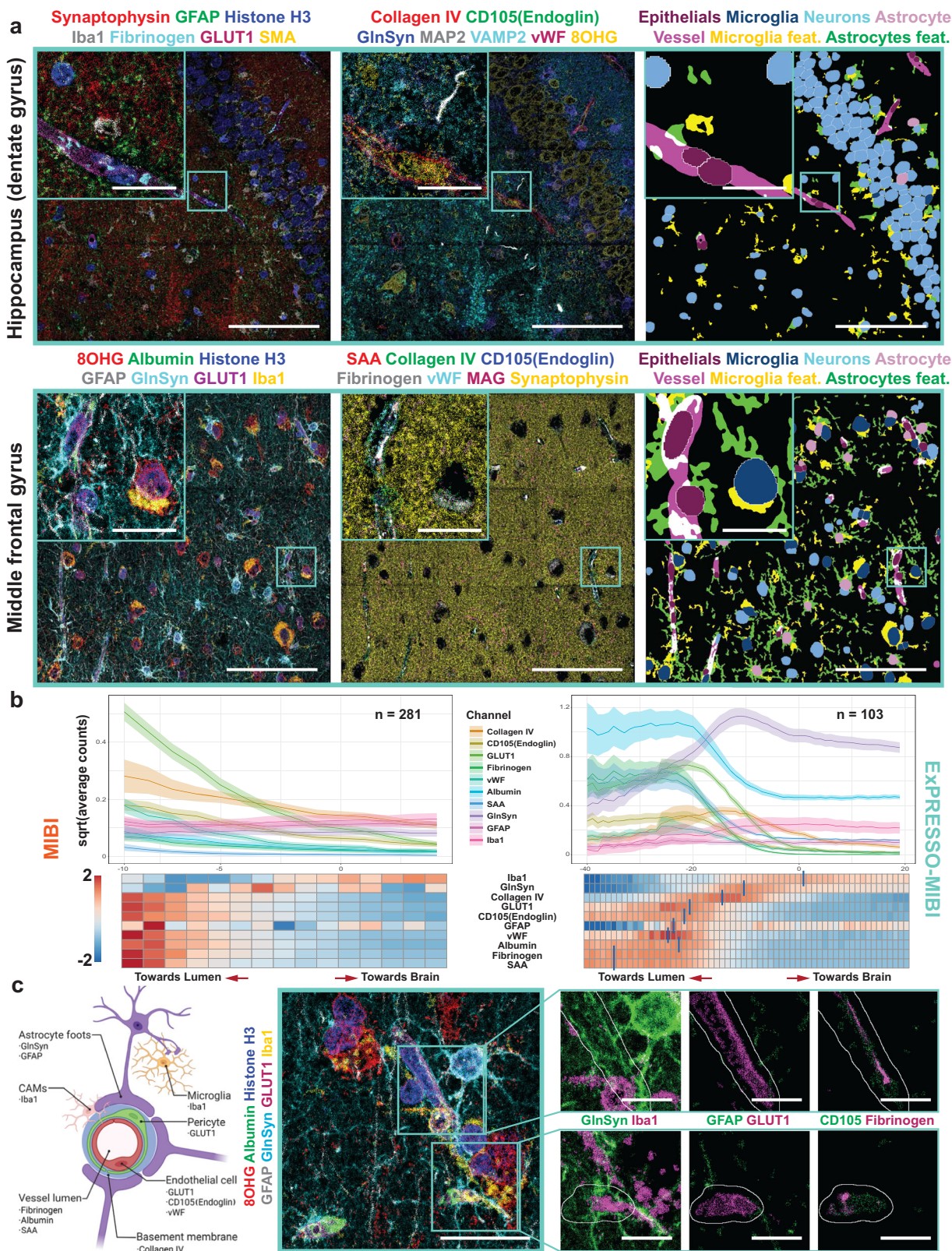

Triton X-100 and incubated 30 min at room temperature with rotation; this was performed three times to remove the SDS.

**Antibody staining.** The gel was first incubated in 1× blocking medium (0.5% BSA, 0.05% (wt/vol) NaN$_3$ in 1× PBS) for 30 min. Meanwhile, the antibody cocktail was prepared by diluting the lanthanide-conjugated

antibodies into antibody diluent (3% normal donkey serum with 0.5% BSA in 1× TBS IHC wash buffer with Tween 20 (Cell Marque, 935B-09)). The gel-embedded tissue section was then incubated with antibody cocktails at 37 °C for 18 h while rotating at 25 RPM. After that, the gel was washed three times with washing buffer (0.1% BSA in 1× TBS IHC wash buffer with Tween 20) for 30 min each wash at 37 °C with rotation.

**Fig. 5 | ExPRESSO enables enhanced interrogation of cells and non-nuclear features, including the BBB, in archival human brain sections. a** MIBI imaging of an ExPRESSO-processed human hippocampus tissue section (top) and middle frontal gyrus (bottom) with tile acquisition of 9 FOVs, each one with a size of 400x400 μm (physical measurements). Shown are markers for pre-synapse (Synaptophysin), astrocyte projections (GFAP), nuclei (Histone H3), microglia (Iba1), vessel lumen and BBB leakiness (Fibrinogen or Albumin or SAA and Fibrinogen), vessels (GLUT1), muscularized vasculature (SMA), basement membrane (Collagen IV), endothelial cells (CD105 and vWF), astrocytes (GlnSyn or GFAP), neurons (MAP2), pre-synapse (VAMP2 or Synaptophysin), neurons with oxidative stress (8OHG), myelin and oligodendrocytes (MAG). Enlarged views of boxed areas are shown as insets. Scale bars (the ExPRESSO-MIBI scale bars indicate the pre- expansion dimensions based on a 3.7-fold expansion): 100 μm (main) and 20 μm (enlarged). **b** Anchoring analysis around vessel feature segment components on both MIBI samples (left) and ExPRESSO-MIBI gels (right) in both curve and heat- map. The average counts were plotted as solid lines, while the 95% confidence intervals were plotted as shadow-around lines. The heatmap of ExPRESSO-MIBI is labeled with blue bars to indicate the local maximum of the respective channel in the ExPRESSO gel analysis. **c** Left: Schematic of components of the BBB and markers used for their identification in the ExPRESSO-MIBI brain panel. Middle and right: Representative complex of BBB, enlarged view from a region in Fig. 5a, and bicolor enlarged regions to further highlight imaging details. Scale bars (the ExPRESSO- MIBI scale bars indicate the pre-expansion dimensions based on a 3.7-fold expan- sion): 250 μm (middle) and 10 μm (right, enlarged).

**Expansion.** The gel with the tissue section was transferred into a 100 mM $NH_4OAc$ solution and rotated for 2 h at room temperature, then expanded in 0.5 mM $NH_4OAc$ until no further expansion was observed (usually three washes of 30 min).

**Adhesion and compress of gel.** The gel was then carefully lifted with a piece of coverslip and placed on top of a glass slide pretreated with Vectabond for IMC or a gold slide for MIBI with the tissue side up. Excessive water was carefully removed with a Kimwipe, then the slide with gel was placed in a sealed dry chamber with Drierite (W. A. Hammond, 23001) and incubated at room temperature overnight or until the gel formed a continuous layer of dry film. The resulting gel can be stored in a desiccated vacuum chamber for 6 months prior to imaging.

### MIBI protocol
The procedure for MIBI staining was previously described[28,34,39]. Briefly, the FFPE block was sectioned onto Vectabond-treated gold slides. After deparaffinization and antigen retrieval process as described in the ExPRESSO protocol, the section was blocked by BBDG (5% normal donkey serum, 0.05% $NaN_3$ in 1× TBS IHC wash buffer with Tween 20), then stained with an antibody cocktail at 4 °C overnight. Subsequently, the sample was washed and post-fixed as described in the ExPRESSO protocol step, before undergoing a series of dehydration steps on the linear stainer (three washes with 100 mM Tris pH 7.5, three washes with ddH₂O, one wash with 70% EtOH, one wash with 80% EtOH, two washes with 95% EtOH, three washes with 100% EtOH, 180 s for each wash), before storage in a vacuum desiccator until acquisition.

### Distortion quantification
Brain sections were expanded and compressed the same as the ExPRESSO Protocol reported in the major manuscript. But specifically, after antigen retrieval, sections were firstly incubated with 1× blocking medium (0.5% BSA, 0.05% (wt/vol) $NaN_3$ in 1× PBS) at RT for 30 min, stained with Hoechst33342 and optional anti-GLUT1 (EPR3915) then anti- rabbit IgG 2nd antibody with Alexa647 (Do not stain fully expanded gel with Hoechst33342 in ddH₂O, which makes the gel fragile and brittle, likely due to multi-positive charges of Hoechst33342). After that, the sections were firstly imaged, then processed as the standard ExPRESSO protocol and stained with the same antibody with the same dilution as the pre-expansion step. At all three gel stages, images were taken with a Keyence BZ-710 fluorescence microscope with Nikon Plan Apo λ ×10/ 0.45 objective at a high-resolution setting. Specifically, to decrease water evaporation during imaging, pre-compression gels were sandwiched between multiple 22 × 50 mm coverslips, while the post-compression gels were simply imaged on the glass slides it compressed to.

Those pairs of regions (for Supplementary Fig. 2e, 5 × 5 mm expanded size, $n = 7$; for Supplementary Fig. 3a, b, 2.4 × 2 mm pre- expansion size) pre- and post-compression were first stitched with MIST[78] as a ImageJ plug-in, then the registration of pre- and post- compression images, distortion test and RMS calculation were performed as previously reported[46,59] with minimal modifications. Briefly, the background of stitched images were first subtracted with ImageJ built-in function (Ball rolling radius = 50 pixels). Elastix[79] was applied to first refine the alignments between pre- and post- compression images (or across remount pre-compression gel) with Transform "AffineTransform" to decrease the distortion caused by sample mounting, then the b-spline nonrigid registration was per- formed to calculate the distortion, a set of features were then selected, their distance changes were measured to give RMS error, with the same codes previously reported[46] with minimal modifica- tions. The line plots of RMS error were then plotted with R package *ggplot2*, where the grayed region depicts ± SD, and the line repre- sents the mean values of RMS error within each step (each group $n = 7$, and each step is a range of 12.15 μm, with a total of 100 steps plotted).

### MIBI and ExPRESSO-MIBI in HeLa cells
HeLa cells (ATCC, CCL-2) were grown in 1× DMEM (Gibco, Invitrogen) with 10% fetal bovine serum, 100 U/mL penicillin (Gibco, Invitrogen), and 100 mg/mL streptomycin (Gibco, Invitrogen), cultured in a cell incubator at 37 °C with 5% $CO_2$ conditions and split with TrypLE Express (Gibco, Invitrogen) every 2–3 days.

Due to the limitation of MIBIscope, if cells were seeded directly on MIBI gold slides, after processing and staining, cytoplasm and bare slides areas around nuclei leads to prevalent blank out and inhomoge- neous etching events, leads to decreased penetration and yield of sig- nals inside nuclei, while the uniform gel-embedded samples in ExPRESSO-MIBI permits adhering cells to be directly imaged, provides much better imaging availability in addition to resolution improvement.

In this scenario, HeLa cells suspension embedded in Histogel were used for the MIBI imaging. HeLa cells were cultured at the same con- dition, then harvested by treating with TrypLE Express (Gibco, Invi- trogen). The resulting cell suspension was centrifuged to form a cell pellet, then fixed as a pellet with 10% PFA in 1 × PBS for overnight (18 hrs) at room temperature (RT). After that, supernatant was removed, liquefied Histogel was added to embed the cell pellet. Once the pellet was solidified, the majority of Histogel was carefully removed, then incubated in 70% Ethanol overnight before a standard histology specimen process. Histogel embedded cell blocks were then sectioned at 4 μm and processed as a standard FFPE section for MIBI staining and imaging.

For ExPRESSO-MIBI, cells were seeded on 12 mm coverslips. After a 1× PBS rinse, cells were fixed with 10% PFA in 1 × PBS for overnight (18 hrs) at room temperature (RT), then permeabilized with 0.5% Triton X-100 in 1× PBS for 15 min at RT, then embedded, denatured, stained, and compressed following the ExPRESSO-MIBI protocol. The cells embedded gels were denatured at 95 °C for 10 to 30 min.

### MIBI-TOF imaging and image processing
Datasets were acquired on a custom MIBI-TOF mass spectrometer equipped with an oxygen duoplasmatron ion gun (Alpha), a custom

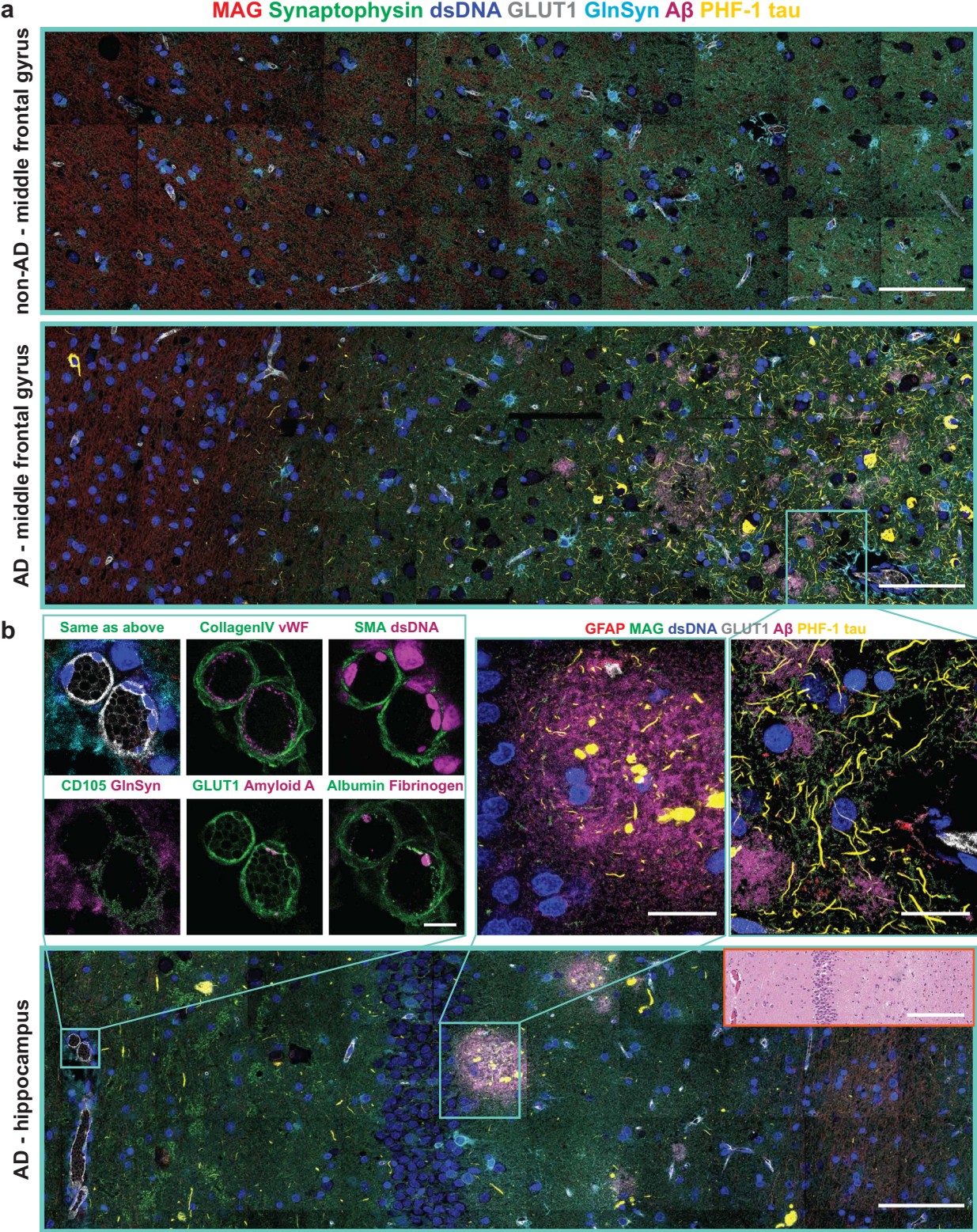

MIBI-TOF mass spectrometer (Betty) equipped with a xenon ion source (Hyperion, Oregon Physics), and a commercially available MIBIscope™ System from Ionpath equipped with a xenon ion source (with MIBI software version v.1.7.0-0f60ffbc). The typical running parameters on instruments are listed below and detailed run conditions related to each experiment are given in Supplementary Table 7.

*Alpha MIBI:*

- Pixel dwell time: 5 ms
- Image area: 400 μm × 400 μm
- Image size: 512 × 512 pixels
- Probe size: 400 nm
- Primary ion current: 3.6 nA on a built-in Faraday cup
- Number of depths: 3

**Fig. 6 | ExPRESSO can be applied to spatially interrogate archival human AD patient samples, to elucidate cellular and extracellular features, including tau tangle, Aβ plaque, and the BBB. a** MIBI imaging of two representative ExPRESSO-processed human middle frontal gyrus tissue sections, one non-AD (top), and one with AD (bottom), respectively. The images shown here includes markers for myelin and oligodendrocytes (MAG), pre-synapse (Synaptophysin), nuclei (dsDNA), vessels (GLUT1), astrocytes (GlnSyn), Aβ plaques (Aβ), and tau tangles (PHF-1 tau). Scale bars (the ExPRESSO-MIBI scale bars indicate the pre-expansion dimensions based on a 3.7-fold expansion): 100 μm (both). **b** MIBI imaging of an ExPRESSO-processed AD-affected human brain section with additional imaging details. Top left: An enlarged view of representative blood vessels from a region in the hippocampus, and dual-colored pseudo images across multiple markers. Top right: An enlarged view of different Aβ plaques and their interaction with tau tangles from a region in the middle frontal gyrus, and spatial features from a region in the hippocampus from the same patient with AD above, images include markers for astrocyte projections (GFAP), myelin and oligodendrocytes (MAG), nuclei (dsDNA), vessels (GLUT1), Aβ plaques (Aβ), and tau tangles (PHF-1 tau). Bottom: a representative region from a large tile of the hippocampus that traverses across the granule cell layer of the dentate gyrus from an AD patient. Markers included at the bottom are the same as **a**. Scale bars (the ExPRESSO-MIBI scale bars indicate the pre-expansion dimensions based on a 3.7-fold expansion): 100 μm (top), 250 μm (top, enlarged), 10 μm (bottom left), and 25 μm (bottom right).

*Betty MIBI:*

- Pixel dwell time: 1 ms
- Image area: 400 μm × 400 μm
- Image size: 1024 × 1024 pixels
- Probe size: 400 nm
- Primary ion current: 5 nA on a built-in Faraday cup
- Number of depths: 1 depth for unexpanded tissues and ExPRESSO tonsil samples, 2 depths for ExPRESSO brain samples

*Production MIBI:*

- Pixel dwell time: 1 ms
- Image area: 400 μm × 400 μm
- Image size: 1024 × 1024 pixels
- Probe size: 400 nm
- Primary ion current: 4.9 nA on a built-in Faraday cup (or the "Fine" imaging mode)
- Number of depths: 1 depth

MIBI image processing, including image extraction and noise removal, was performed using MIBI Analysis tools according to package manuals (https://github.com/lkeren/MIBIAnalysis) as previously described[28,80]. For MIBI images that were acquired over multiple Z-depths, samples were aligned and summed using the Ki-67 (tonsil) or Histone H3 (brain) channels. Imaging stitching was performed with custom MatLab scripts initially developed by Dmitry Tebaykin (https://github.com/dtebaykin/MibiStitch) and flat-field correction by Sizun Jiang.

### IMC acquisition and image processing
ExPRESSO samples and FFPE tissue sections on glass slides were ablated after staining using the Hyperion Imaging Mass Cytometry (Fluidigm). Before data acquisition, the Hyperion was auto-tuned using a three-element tuning slide (Fluidigm). Regions of interest were ablated at a frequency of 200 Hz with an ablation energy of 0 and 1 dB for the ExPRESSO samples and FFPE tissue sections, respectively. MCD and txt files were exported and visualized with the MCD viewer from Fluidigm. MCD files were converted to single-marker tiff images using a custom Python script developed by the Bodenmiller group (https://github.com/BodenmillerGroup/imctools), and denoising was performed as described in the MIBI-TOF image processing above.

### Image segmentation
Cell segmentation was performed with a local implementation of deepcell-tf 0.6.0 or 0.9.1 as described[47,81]. Cell segmentation in the tonsil images was performed using Histone H3 for the nucleus and CD45 for membrane features. For the brain images (hippocampus and middle frontal gyrus), Histone H3 was used for the nucleus channel and a dummy (all-zero) image was used as the membrane channel. Signals from these channels were first capped at the 99.7th percentile before input into the model. The deepcell-tf version used to generate the final segmentation mask. Parameters for *model_mpp* for cell segmentation are summarized in Supplementary Table 9.

### Tonsil phenotyping
Marker expression levels for each single-cell from MIBI and IMC images were extracted, log1p transformed, and z-normalized, with the R functions *log1p()* and *scale()*. The R package *Seurat* (3.2.3) was then used for unsupervised clustering and cell phenotyping. The markers CD3, CD20, Cytokeratin, MPO, CD38, and CD11c were used for the MIBI and ExPRESSO-MIBI dataset, while CD68, CD20, CD4, and CD8 were used for the IMC and ExPRESSO-IMC dataset. Specifically, *FindNeighbors(dims = 1:15)* and *FindClusters(res = 0.5)* from *Seurat* were used for heatmap generation, which was followed by manual annotation of each cluster.

### Brain cell phenotyping
Cells were clustered with the R package *FlowSOM* (2.4.0)[82] based on the following markers: Iba1 (microglia), HLA-DR (microglia), GFAP and GlnSyn (astrocyte), CD105, Fibrinogen, SMA, GLUT1, vWF, Albumin, and CD36 (endothelial related). Thirty meta clusters were extracted and merged into four main categories.

### Feature segment of non-nuclear components
Non-nuclear components such as vessels were segmented with the ez_segment_gui.m, which was described in Vijayaragavan, et al., 2022, the package is available from the MBI Analysis User Interface (https://github.com/angelolab/MAUI)[80]. Non-nuclear components were identified using composite channels of GLUT1, vWF, and Fibrinogen (vessels), GlnSyn and GFAP (astrocytes), and Iba1 and HLA-DR (microglia).

### MIBI image analysis
**Related to Fig. 1e: expansion fold quantification.** Linear: The distance between two different features on the same piece of tissue pre- and post- ExPRESSO was measured and the ratio was calculated. Area: Nuclear staining (with Hoechst33342) of four FFPE tonsils were imaged pre- and post- ExPRESSO, the areas of tissues were measured in ImageJ, and the ratio was calculated. Segment: Four 1200 × 1200 μm tiles of ExPRESSO-MIBI tonsil images and four 1200 × 1200 μm tiles of MIBI tonsil images were segmented as described in the Image Segmentation section. The cell size distribution ratio was calculated.

**Related to Fig. 1f: long-term storage.** Different but nearby regions of the same piece of ExPRESSO-MIBI tonsil gel were imaged on 2021-07-27, 2021-09-17, and 2022-02-18 under the same imaging parameters. Histone H3 and dsDNA signals of five different fields of view (FOVs) from each batch were summed and plotted.

**Related to Fig. 1g: current counts.** The same piece of ExPRESSO tonsil gel was imaged on MIBI with the same FOV size and resolution but different currents (as described in Supplementary Table 7). The Histone H3 and dsDNA signals were averaged across two or three runs for each condition.

**Related to Fig. 1h: current counts.** A MIBI tonsil sample and a parallel ExPRESSO-MIBI tonsil gel were imaged with the same ion dose density (as described in Supplementary Table 7), but the grid size of the

ExPRESSO sample was four times that of the MIBI (Supplementary Fig. 4c). Average Histone H3 counts from three runs are shown for each condition.

**Related to Supplementary Fig. 3c: resolution.** Line scans to calculate the resolution based on the 16–84% criteria were applied on the 115-Histone H3 MIBI and ExPRESSO-MIBI images from FFPE human tonsil, as previously reported[83]. Five line scans were measured under each conditions and the apparent resolution in physical distance of both conditions were calculated. The values are reported as average ± s.d.

**Related to Fig. 3a/Supplementary Fig. 8a: line scan.** For both unexpanded and ExPRESSO, images after background removal but without denoising were used. The line profiles were generated with ImageJ. The ExPRESSO images were scaled down by expansion fold to match the unexpanded images.

**Related to Supplementary Fig. 10b: brain marker gradients.** For the three selected channels (CD56, MAG, and GFAP), the signals of the 1x16 stitched images were summed perpendicular to the long axis to get the expression level changes along the long axis. Then the expression level for each protein were Min-Max normalized within each channel, where Min is the 5% quantile and Max is the 95% quantile value for the individual markers. Subsequently, each 'step' was defined as 200 adjacent vertical pixel lines with the MIBI scans across regions. Each 'step' slides 50 pixels lines in horizontal direction relative to the previous 'step' (e.g., step 1: vertical row #1-200, step 2: vertical row #50–250). The normalized average value of individual marker expression levels in each 'step' were then plotted, along with the 95% confidence intervals.

**Related to Fig. 5b/Supplementary Fig. 12b: brain vessel anchoring analysis.** The brain vessel object was identified as described in the previous section and then the microenvironment around each individual vessel was extracted layer-by-layer through stepwise morphology operation on the vessel feature segment objects. Each 'step' area was defined as the 1 pixel-width extension/inclusion area around the vessel mask, which was generated by subtracting the results from MATLAB functions: *imdilate()* and *imerode()*.Specifically, in the ExPRESSO dataset of both non-AD and AD cases, all channels were first normalized by the median of dsDNA channel across each tile to prevent cross-tile signal fluctuations due to instrumental factors. To avoid interference from neighboring vessels during the area extensioning, the areas of each 'step' that overlapped with other non-dilated vessel objects were removed. Subsequently, the marker expression level, cell phenotype fraction (number of pixels for certain cell types in fraction of the total pixels numbers in each 'step' area), and non-nuclear feature segment object fraction (number of pixels for certain non-nuclear feature segment object in fraction of the total pixels numbers in each 'step') were extracted.

For downstream analysis, values from each step area of each individual vessel were square-root transformed for visualization purposes. In the line plot, average counts and 95% confidence intervals were plotted (EXPRESSO-MIBI samples: −40 to +20 pixels; MIBI sample: −10 to +5 pixels). For the heatmap of marker expressions along the 'steps' of brain vessels, the expression levels for each marker were Z-normalized within all the 'steps' (EXPRESSO-MIBI samples: −40 to +20 pixels; MIBI sample: −10 to +5 pixels), then plotted with R function *heatmap.2()*. The line indicating the maximum value for each channel was found by R function *max()*.

## Data visualization

Single-channel and multi-color images were assembled with ImageJ. Scale bars of ExPRESSO samples were based on instrumental FOV size, then back-calculated with the 3.7-fold expansion magnification to indicate pre-expansion dimensions. Visualizations of the analysis results were either produced using Excel, or R packages *ggplot2* and *pheatmap*.

## Statistics and reproducibility

For reproducibility, all experiments were repeated independently at least three times, unless otherwise specified.

## Reporting summary

Further information on research design is available in the Nature Portfolio Reporting Summary linked to this article.

## Data availability

All the multiplexed imaging data (MIBI and IMC), and other images used in the Figures of this work, have been deposited in the Zenodo under accession code DOI:10.5281/zenodo.7960511[84].

## Code availability

The codes for anchoring analysis are made available at https://github.com/yunhaoBai/Anchoring_analysis/tree/0.1.0[85].

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

## Acknowledgements

The authors thank previous and current members of Ionpath Inc, including Matthew Newgren and Maciej Zerkowski, for their unwavering technical support. We thank members of the Nolan, Angelo, Bendall, and Jiang labs for helpful discussions. We also thank Professor Yongxin Zhao for insightful discussions related to this project. We are grateful to Pauline Chu for her technical assistance in tissue sectioning. We thank Thomas Carver from the Microfab Shop of SNSF for his thin-film evaporation services. We also thank Chuck Hitzman for his helpful discussion and guidance on SIMS methodology and experiments. We thank Syed A Bukhari from the Thomas Montine laboratory for providing the archival human brain tissues. Figs. 1a, 4c, 5c, and Supplementary Figs. 1a contain components created with BioRender.com. This study was supported by the Leukemia & Lymphoma Society Career Development Program (S.J.), Stanford Dean's Fellowship (S.J.), EMBO postdoctoral fellowship ALTF 300-2017 (X.R.-C), Stanford Graduate Fellowship (B.Z. and N.G.), Stanford Maternal and Child Health Research Institute Postdoctoral Support Award (D.F), T32 AI007290 (B.J.C), NCI CA246880-01 (N.G.), Canadian Institutes of Health Research Postdoctoral Fellowship (J.P.O.), Banting Postdoctoral Fellowship (J.P.O.), K08 AG065502 (M.B.M.), the Brigham and Women's Hospital Program for Interdisciplinary Neuroscience through a gift from L. and T. Rand (M.B.M.), the donors of the Alzheimer's Disease Research program of the BrightFocus Foundation A20201292F (M.B.M.), the Doris Duke Charitable Foundation Clinical Scientist Development Award 2021183 (M.B.M.), NIH R01 AG056287, R01 AG057915, R01 AG068279, and U19 AG065156 (M.A. and S.B.), R35 GM137936, P01 HD106414 (B.G.), Bill & Melinda Gates Foundation INV-002704 (S.J.), OPP1113682 (G.P.N.), US Food and Drug Administration Medical Countermeasures Initiative contracts HHSF223201610018C and 75F40120C00176 (G.P.N.), Parker Institute for Cancer Immunotherapy (G.P.N.), a Rachford and Carlota A. Harris Endowed Professorship (G.P.N.), NIH DP2AI171139 (S.J.) and the Gilead Research Scholar in Hematologic Malignancies (S.J.). This article reflects the views of the authors and should not be construed as representing the views or policies of the FDA, NIH, BMGF, or other institutions that provided funding.

## Author contributions

Y.B., X.R.-C., and S.J. conceived of the presented idea and planned the experiments. Y.B. developed the methodology. Y.B., B.Z., B.J.C., and S.J. developed the software used in data analysis. Y.B., B.Z., J.P.O., B.J.C., K.V., S.C.B., X.R.-C., and S.J. participated in the formal analysis of the datasets. Y.B., B.Z. analyzed and visualized the data. J.P.O., D.F., M.B., K.V., B.J.C., N.G., D.P., C.M.S., E.A.G., B.G., M.B.M., S.M.N., S.J.R., M.A., S.C.B. contributed to tools, reagents and experimental materials. Y.B. wrote the manuscript with support from B.Z., X.R.-C., and S.J. X.R.-C., G.P.N., and S.J. supervised the project. All authors reviewed and edited the final manuscript.

## Competing interests

G.P.N., S.C.B., and M.A. are co-founders and stockholders of Ionpath Inc, which manufactures the instrument used in this manuscript. G.P.N. is a co-founder and stockholder of Akoya Biosciences, Inc. and an inventor on patent US9909167. G.P.N. is a Scientific Advisory Board member for Akoya Biosciences, Inc. G.P.N. received research grants from Pfizer, Inc.; Vaxart, Inc.; Celgene, Inc.; and Juno Therapeutics, Inc. during the time of this work. C.M.S. is a scientific advisor to, has stock options in, and has received research funding from Enable Medicine, Inc; S.J.R. has received research support from Affimed, Merck, and Bristol-Myers Squibb (BMS) and is a member of the Scientific Advisory Board for Immunitas Therapeutics, all outside this work. The Board of Trustees of the Leland Stanford Junior University has a pending US patent with the named

inventors Y.B., B.Z., X.R.-C., G.P.N., and S.J., on the "Preparation and Imaging of an Expanded and Compressed Vacuum-Stable Gels for Multiplexed High-Resolution Spatial Interrogation of Biospecimens Proteome". The other authors declare no competing interests.
