## [Peer Review File · Nature Communications]

Expanded Vacuum-Stable Gels for Multiplexed High Resolution Spatial HistopathologyThis manuscript has been previously reviewed at another journal that is not operating a transparent peer review scheme. This document only contains reviewer comments and rebuttal letters for versions considered at *Nature Communications*.

REVIEWERS' COMMENTS

Reviewer #1 (Remarks to the Author)

Bai et al have presented a revised paper on a novel approach for expansion imaging that involves spatially-resolved mass analysis. The authors have demonstrated a commendable level of responsiveness by providing thorough and detailed responses to the multiple requests made by the reviewers and editors.

While it may be a minor detail, I still hold a different view from the authors with regards to the reversibility of crosslinks through the use of a detergent. Although the authors have made efforts to find publications demonstrating the contrary, covalent bonds between proteins are generally difficult to reverse due to the strong and stable bond created by the sharing of electrons between atoms. Breaking a covalent bond typically requires a significant amount of energy, which makes it challenging to reverse covalent bonds formed through crosslinking with paraformaldehyde or other crosslinking agents. Nonetheless, I appreciate the authors' perspective on this matter and commend their efforts to address it in the paper.

My major concern is, as Reviewer 3, I remain unconvinced that the EXPRESSO technique is suitable for depth profiling and 3D reconstruction. The authors have not provided sufficient evidence to demonstrate the following:

- Homogeneity in the expansion process: The authors need to show that the expansion of tissue is sufficiently homogeneous across the sample, as non-uniform expansion could lead to errors in depth profiling and 3D reconstruction.
- Absence of background noise: It is important to ensure that the compression in the Z plane does not result in the fusion or superposition of certain structures, which could create background noise in the images. At this spatial resolution, it could be a critical point as in the case of fluorescence microscopy, where confocal analysis is necessary to obtain the signal from a single plane to ensure that the signal is specific to the region of interest.
- Homogeneous depth ablation: The authors need to demonstrate that the depth ablation is homogeneous across different tissue structures. The presence of signal loss may indicate uneven ablation, which could lead to inaccuracies in the 3D reconstruction.

By addressing these concerns, the authors can provide a more comprehensive assessment of the suitability of EXPRESSO for depth profiling and 3D reconstruction.

I would advise the authors to consider removing this section, as it is not the main focus of the paper, and instead focus on addressing these points in a separate publication.

The last point to consider is that interferences may arise due to the presence of the gel. As demonstrated in Figure R2 on Page 24, the authors noted signal losses, but were unable to provide a precise explanation. This effect is also observed in other mass spectrometry imaging techniques, such as SIMS or MALDI-axialTOF. These signal losses represent an extreme manifestation of an ionization problem that results from the diversity of structures. This phenomenon can be less pronounced, leading to differences in signal intensity during analysis, independent of the protein concentration. Consequently, I remain skeptical about the quantitative accuracy of the results.

Reviewer #2 (Remarks to the Author)

In their rebuttal/cover letter, the authors address the two major points I raised in the second round of prior review.

1. With respect to signal intensity after expansion, they comment on various aspects of the protocol that may increase access of antibodies to epitopes and limit the predicted loss of signal due to volume dilution upon expansion. These are reasonable and may be correct, but at this point the key is to make the reader very aware of this matter because if the epitopes of biological concern are among those where signal is substantially lost, then this becomes a severe limiting feature of the method. Perusing the list provided by the authors as Table R1, many although not all of the data fit into a pretty clear pattern. Staining for epitopes associated with structural elements such as nucleic acids and extracellular matrix show less than predicted loss of signal and even a gain relative to non-EXPRESSO, whereas those associated with membrane proteins (CD3

being a major exception), lose signal by several fold. This makes sense as the epitopes on macromolecular repetitious structures like matrix and nucleic acids would benefit from 'unpacking' whereas sparse membrane proteins would not and they would suffer from dilution as expected, albeit less than theoretically maximal. I think this table should be incorporated into the final manuscript and the authors should include a discussion on the issue citing to these data, which can provide a guide to users about whether the markers of interest in their study are or are not likely to lose signal when the sample is processed by ExPRESSO and they can then take into account S/N loss vs. resolution gain in determining whether to adopt the method.

2. With respect to which antibodies work and which do not, it is obviously not possible or practical for the authors to provide a comprehensive guide to users on this issue, as the basis for failure in this method vs. others is not trivially apparent. Relying on information in the citations, the accumulating database on antibodies for MIBI use, and direct in lab testing will be how users can most efficiently proceed when establishing a panel for a specific purpose.

With the changes I suggested in point 1, I can recommend this paper to NC, pending satisfactory responses to the points raised by other referees.

Reviewer #3 (Remarks to the Author)

The reviewer does not have any further concerns regarding the current version of the manuscript, which has been adequately addressed and is now suitable for publication in Nature Communications.

RESPONSE TO REVIEWERS' COMMENTS

We are grateful to the Reviewers for assessing this work and for their advice and positive feedback. In our revised manuscript, we have:

1. Removed the section on depth profiling, as suggested by Reviewer #1.
2. Included Supplementary Table 5, as suggested by Reviewer #2.

Expanded upon the Discussion to better clarify the points highlighted by Reviewers #1 and #2.

Reviewer #1:

Remarks to the Author:

Bai et al have presented a revised paper on a novel approach for expansion imaging that involves spatially-resolved mass analysis. The authors have demonstrated a commendable level of responsiveness by providing thorough and detailed responses to the multiple requests made by the reviewers and editors.

We thank the Reviewer for highlighting our efforts.

While it may be a minor detail, I still hold a different view from the authors with regards to the reversibility of crosslinks through the use of a detergent. Although the authors have made efforts to find publications demonstrating the contrary, covalent bonds between proteins are generally difficult to reverse due to the strong and stable bond created by the sharing of electrons between atoms. Breaking a covalent bond typically requires a significant amount of energy, which makes it challenging to reverse covalent bonds formed through crosslinking with paraformaldehyde or other crosslinking agents. Nonetheless, I appreciate the authors' perspective on this matter and commend their efforts to address it in the paper.

We thank the Reviewer for this opinion. We would like to clarify that the reverse crosslinking is primarily achieved by the heat-induced epitope retrieval (HIER) step, not the later non-enzymatic, heat and detergent-based treatment (which is generally termed "denaturation" of the hydrogel-embedded sample). These sequential steps are schematically presented in Figure 1a and described in detail in the Material and Methods section.

While the exact mechanism of HIER is still under debate, **empirical scientific evidence** supported by virtually all histology labs in the world show that the chemical modifications introduced by PFA within the tissues are at least partially reversed, to the point that it is practical to perform antibody or *in situ* probe staining which otherwise fail.

It is outside of the scope of this paper to determine the chemical mechanism, but our empirical scientific evidence also shows that the HIER step also allows for the robust expansion and antibody staining data we present here.

My major concern is, as Reviewer 3, I remain unconvinced that the EXPRESSO technique is suitable for depth profiling and 3D reconstruction. The authors have not provided sufficient evidence to demonstrate the following:

- Homogeneity in the expansion process: The authors need to show that the expansion of tissue is sufficiently homogeneous across the sample, as non-uniform expansion could lead to errors in depth profiling and 3D reconstruction.

- Absence of background noise: It is important to ensure that the compression in the Z plane does not result in the fusion or superposition of certain structures, which could create background noise in the images. At this spatial resolution, it could be a critical point as in the case of fluorescence microscopy, where confocal analysis is necessary to obtain the signal from a single plane to ensure that the signal is specific to the region of interest.

- Homogeneous depth ablation: The authors need to demonstrate that the depth ablation is homogeneous across different tissue structures. The presence of signal loss may indicate uneven ablation, which could lead to inaccuracies in the 3D reconstruction.

By addressing these concerns, the authors can provide a more comprehensive assessment of the suitability of EXPRESSO for depth profiling and 3D reconstruction.

I would advise the authors to consider removing this section, as it is not the main focus of the paper, and instead focus on addressing these points in a separate publication.

The Reviewer raises a set of valid points. We would like to point out the following evidence:

(1) We have demonstrate that tissue expansion is sufficiently homogeneous across the sample (Figure S1g to i);

(2) As addressed in point 5 of the first round of revision, relative spatial position is conserved after compression (Figure S5, Oran et al., Science 2018). Regarding the comparison with fluorescence microscopy, the Reviewer is likely aware that the principles of fluorescence imaging are different from mass spectrometry imaging (MSI). In MSI, only the topmost layer is ablated and the following imaging process further removes the substrates layer by layer which make its axial resolution much higher than the lateral, a major difference to fluorescence imaging. Moreover, fluorescence microscopy is a valid technology that has provided extensive biological knowledge, as such, even in the event of increased superposition, ExPRESSO and its high multiplexed and resolution capabilities would be a relevant tool for the life sciences, especially because the local molecular environments will share crowding features;

(3) There should be reduced tissue heterogeneity in ExPRESSO samples compared with non-expanded samples. Hydrogel embedding provides a homogeneous sample support ameliorating topographical heterogeneity (Figure S1m), which reduces (if not removes) depth ablation issues compared with a non-expanded sample in normal MSI.

Still, this comment was helpful to reassess the main conclusions we want to share in the paper, and after internal consideration, we have decided to follow the Reviewer's advice on removing this section from the Result and Supplementary Figure of the current manuscript. Given the discussions from other peer-reviewed papers on this topic, we have maintained the following sentence in the Discussion section of the manuscript (page 12, line 540 to 548) to highlight future directions of this technology:

“Although the compressing step allows the ExPRESSO gels to withstand high-vacuum environments with an apparent sacrifice of axial resolution, MIBI and many other analytical methods have superior axial resolution over lateral resolutions with the 5 to 100 nm range (Rovira-Clave et al, Nature Communication 2021; Keren et al., Science advances 2019), allowing it to recapture axial resolutions, opening the way to 3D depth profiling in thick tissue sections, although this would require extensive further validation experiments.”

Oran, D., Rodrigues, S. G., Gao, R., Asano, S., Skylar-Scott, M. A., Chen, F., ... & Boyden, E. S. (2018). 3D nanofabrication by volumetric deposition and controlled shrinkage of patterned scaffolds. *Science*, 362(6420), 1281-1285.

The last point to consider is that interferences may arise due to the presence of the gel. As demonstrated in Figure R2 on Page 24, the authors noted signal losses, but were unable to provide a precise explanation. This effect is also observed in other mass spectrometry imaging techniques, such as SIMS or MALDI-axialTOF. These signal losses represent an extreme manifestation of an ionization problem that results from the diversity of structures. This phenomenon can be less pronounced, leading to differences in signal intensity during analysis, independent of the protein concentration. Consequently, I remain skeptical about the quantitative accuracy of the results.

We thank the Reviewer for this comment. The signal losses referred to by the Reviewer are random and rare events that represent <0.1% of the regions imaged, typically explained by ionization suppression due to sample charge buildup. These signal losses are not necessarily related to local structures, especially when the ExPRESSO samples are enclosed in one piece of homogeneous gel. In our hands, this effect is equally observed in **BOTH** ExPRESSO and conventional MIBI (unexpanded tissue samples), as demonstrated in Figure R1 (which is also Figure R2 in the 1st round of revision, reproduced as Figure R1 below for convenience). Therefore, this issue has minimal consequences and it is not related to the ExPRESSO technology, but a widespread drawback of the mass spectrometry imaging.

Figure R1, blank-out event in an ExPRESSO gel, indicated by VAMP2 channel (left); blank-out event in a piece of unexpanded tissue, indicated by Histone H3 channel (right). Scale bar (both physical size): 100 μm (both).

Regarding the quantitative accuracy of the results, it has been carefully demonstrated previously that MIBI is a quantitative technology (Keren et al., Science advances 2019). Moreover, in this ExPRESSO manuscript, we use quantitative **relative** protein enrichment for cell type identification (Figure 3e, S3e, 5a, and S5a), **relative** spatial protein positioning for non-nuclear feature identification (Figure 4c, and 5a), and the “walking” computational approach **in relationship to** the Blood Brain Barrier to discern multilayered structures (Figure 5b, S5c, and S5e). Thus, our results hold true empirically. To better clarify this point for readers, we have now expanded on this topic on the Discussion section of the manuscript (page 12, line 510 to 517):

“In SIMS, signal intensity is not only depending on the enrichment of the readout element but also affected by the local composition of the sample. Still, protein level quantification is robust by MIBI (Keren et al., Science advances 2019), and the multiparameter and high resolution capabilities of ExPRESSO enable harnessing relative expression and spatial location for quantitative and qualitative description of the samples of interest.”

Reviewer #2:

Remarks to the Author:

In their rebuttal/cover letter, the authors address the two major points I raised in the second round of prior review.

1. With respect to signal intensity after expansion, they comment on various aspects of the protocol that may increase access of antibodies to epitopes and limit the predicted loss of signal due to volume dilution upon expansion. These are reasonable and may be correct, but at this point the key is to make the reader very aware of this matter because if the epitopes of biological concern are among those where signal is substantially lost, then this becomes a severe limiting feature of the method. Perusing the list provided by the authors as Table R1, many although not all of the data fit into a pretty clear pattern. Staining for epitopes associated with structural elements such as nucleic acids and extracellular matrix show less than predicted loss of signal and even a gain relative to non-EXPRESSO, whereas those associated with membrane proteins (CD3 being a major exception), lose signal by several fold. This makes sense as the epitopes on macromolecular repetitious structures like matrix and nucleic acids would benefit from 'unpacking' whereas sparse membrane proteins would not and they would suffer from dilution as expected, albeit less than theoretically maximal. I think this table should be incorporated into the final manuscript and the authors should include a discussion on the issue citing to these data, which can provide a guide to users about whether the markers of interest in their study are or are not likely to loss signal when the sample is processed by EXPRESSO and they can then take into account S/N loss vs. resolution gain in determining whether to adopt the method.

The Reviewer is right that further discussing this topic would be beneficial for future users of the EXPRESSO technology. We have now incorporated this table into the manuscript as recommended (Supplementary Table 5) and modified the Discussion to better clarify this point (page 12, lines 498 to 510),

from:

“Despite theoretical signal dilution, quantification of all markers was higher than expected in EXPRESSO, suggestive of additional factors in play such as increased epitope accessibility by molecular decrowding (Sarkar et al., 2022).”

To:

“Despite theoretical signal dilution, quantification of all markers was higher than expected in the EXPRESSO tonsil (Supplementary Table 5) and in particular 8 out of 17 antibodies had a higher count per pixel over the same non-expanded area, suggestive of additional factors in play such as increased epitope accessibility by molecular decrowding (Sarkar et al.,

2022). Still, certain membrane proteins (e.g., CD20, CD11b, CD45), which tend to be sparsely distributed, had lower counts per pixel compared to the non-expanded samples (although the overall counts are higher than expected considering dilution by expansion), which highlights the need for extensive titration and careful validation of antibody panels for ExPRESSO.”

2. With respect to which antibodies work and which do not, it is obviously not possible or practical for the authors to provide a comprehensive guide to users on this issue, as the basis for failure in this method vs. others is not trivially apparent. Relying on information in the citations, the accumulating database on antibodies for MIBI use, and direct in lab testing will be how users can most efficiently proceed when establishing a panel for a specific purpose.

We thank the Reviewer for the kind words of support.

With the changes I suggested in point 1, I can recommend this paper to NC, pending satisfactory responses to the points raised by other referees.

We thank the Reviewer for the guidance during the revision process, which certainly improved the manuscript.

Reviewer #3:

Remarks to the Author:

The reviewer does not have any further concerns regarding the current version of the manuscript, which has been adequately addressed and is now suitable for publication in Nature Communications.

We thank the Reviewer for the insightful feedback during the revision process.

REVIEWERS' COMMENTS

Reviewer #1 (Remarks to the Author):

I have reviewed the submitted manuscript and I have no further comments to add. The authors have done an excellent job in addressing all the previous comments that were raised, and I am satisfied with the revisions made. The manuscript is now suitable for publication in its current form.